EMBO
Molecular Medicine

# Functional systemic CD4 immunity is required for clinical responses to PD-L1/PD-1 blockade therapy

Miren Zuazo[1,†], Hugo Arasanz[1,†], Gonzalo Fernández-Hinojal[2,†], Maria Jesus García-Granda[1,†], María Gato[1,†], Ana Bocanegra[1], Maite Martínez[2], Berta Hernández[2], Lucía Teijeira[2], Idoia Morilla[2], Maria Jose Lecumberri[2], Angela Fernández de Lascoiti[2], Ruth Vera[2,*] [iD], Grazyna Kochan[1,**] [iD] & David Escors[1,3,***] [iD]

## Abstract

The majority of lung cancer patients progressing from conventional therapies are refractory to PD-L1/PD-1 blockade monotherapy. Here, we show that baseline systemic CD4 immunity is a differential factor for clinical responses. Patients with functional systemic CD4 T cells included all objective responders and could be identified before the start of therapy by having a high proportion of memory CD4 T cells. In these patients, CD4 T cells possessed significant proliferative capacities, low co-expression of PD-1/LAG-3 and were responsive to PD-1 blockade *ex vivo* and *in vivo*. In contrast, patients with dysfunctional systemic CD4 immunity did not respond even though they had lung cancer-specific T cells. Although proficient in cytokine production, CD4 T cells in these patients proliferated very poorly, strongly co-upregulated PD-1/LAG-3, and were largely refractory to PD-1 monoblockade. CD8 immunity only recovered in patients with functional CD4 immunity. T-cell proliferative dysfunctionality could be reverted by PD-1/LAG-3 co-blockade. Patients with functional CD4 immunity and PD-L1 tumor positivity exhibited response rates of 70%, highlighting the contribution of CD4 immunity for efficacious PD-L1/PD-1 blockade therapy.

**Keywords** B7-H1; biomarker; immunotherapy; lung cancer; PD-1/PD-L1
**Subject Categories** Cancer; Immunology

## Introduction

PD-L1/PD-1 blockade is demonstrating remarkable clinical outcomes since its first clinical application in human therapy (Brahmer *et al*, 2012; Topalian *et al*, 2012). These therapies interfere with immunosuppressive PD-L1/PD-1 interactions by systemic administration of blocking antibodies. PD-L1 is overexpressed by many tumor types and generally correlates with progression and resistance to pro-apoptotic stimuli (Azuma *et al*, 2008; Gato-Canas *et al*, 2017; Juneja *et al*, 2017). PD-1 is expressed in antigen-experienced T cells and interferes with T-cell activation when engaged with PD-L1 (Chemnitz *et al*, 2004; Karwacz *et al*, 2011). The majority of advanced non-small-cell lung cancer (NSCLC) patients progressing from conventional cytotoxic therapies who receive PD-L1/PD-1 blockade therapy do not respond. The causes for these distinct clinical outcomes are a subject for intense research (Topalian *et al*, 2016). Emerging studies indicate that PD-L1/PD-1 blockade therapy does not only affect the tumor microenvironment, but also alters the systemic dynamics of immune cell populations (Hui *et al*, 2017; Kamphorst *et al*, 2017a,b; Krieg *et al*, 2018). Some of these changes do correlate with responses and could be used for real-time monitoring of therapeutic efficacy. For example, PD-1[+] CD8 T cells expand systemically after PD-1 blockade therapy in lung cancer patients (Kamphorst *et al*, 2017a). As CD8 T cells are the main direct effectors of responses through cytotoxicity over cancer cells, these changes are thought to be the consequence of efficacious anti-tumor immunity. Indeed, CD8 T-cell infiltration of tumors correlates with good outcomes (Daud *et al*, 2016). However, the role of CD4 immunity in patients undergoing PD-L1/PD-1 blockade therapy remains poorly understood although extensive pre-clinical data link CD4 responses to anti-tumor immunity. Hence, CD4 T cells recognizing tumor neoepitopes contribute significantly to the efficacy of several types of immunotherapies in murine models and in cancer patients (Kreiter *et al*, 2015; Knocke *et al*, 2016; Sahin *et al*, 2017).

Human T cells undergo a natural differentiation process following the initial antigen recognition, characterized by the progressive loss of CD27 and CD28 surface expression, and acquisition of

1  Immunomodulation Group, Biomedical Research Center of Navarre-Navarrabiomed, Fundación Miguel Servet, IdISNA, Pamplona, Spain
2  Department of Oncology, Hospital Complex of Navarre, IdISNA, Pamplona, Spain
3  Division of Infection and Immunity, University College London, London, UK
   *Corresponding author. Tel: +34 848 422162; E-mail: ruth.vera.garcia@navarra.es
   **Corresponding author. Tel: +34 848 425742; E-mail: grazyna.kochan@navarra.es
   ***Corresponding author. Tel: +34 848 425742; E-mails: d.escors@ucl.ac.uk; descorsm@navarra.es
   † These authors contributed equally to this work

memory and effector functions (Lanna *et al*, 2014, 2017). Hence, human T cells can be classified according to their CD27/CD28 expression profiles into poorly differentiated (CD27$^+$ CD28$^+$), intermediately differentiated (CD27$^{negative}$ CD28$^+$), and highly differentiated (CD27$^{negative}$ CD28$^{low/negative}$, $T_{HD}$) subsets (Lanna *et al*, 2014). Highly differentiated T cells in humans are composed of memory, effector, and senescent T cells, all of which could modulate anti-cancer immunity in patients and alter susceptibility to immune checkpoint inhibitors. To understand the impact of systemic CD4 and CD8 T-cell immunity before the start of immunotherapies, we carried out a discovery study in a cohort sample of 51 NSCLC patients undergoing PD-1/PD-L1 immune checkpoint blockade therapy after progression to platinum-based chemotherapy. Our results indicate that baseline functional systemic CD4 immunity is required for objective clinical responses to PD-L1/PD-1 blockade therapies.

# Results

## The baseline percentage of systemic CD4 $T_{HD}$ cells within CD4 cells separates NSCLC patients into two groups with distinct clinical outcomes

To study whether there was a correlation between specific systemic T-cell subsets and responses to anti-PD-L1/PD-1 immunotherapy in NSCLC patients, a prospective study was carried out in a cohort of 51 patients treated with PD-L1/PD-1 inhibitors (Table EV1). These patients had all progressed to conventional cytotoxic therapies and received immunotherapies as part of their treatments. 78.4% presented an ECOG of 0–1, 70.6% with at least three affected organs, and 25.5% with liver metastases (Table EV1).

First, the percentages of CD4 T-cell differentiation subsets according to CD27/CD28 expression profiles were quantified within total CD4 cells in patients before the start of immunotherapies (baseline) from fresh peripheral blood samples and compared to healthy age-matched donors. Overall, cancer patients showed a significantly higher baseline percentage of CD4 $T_{HD}$ cells than healthy controls ($P < 0.001$; Fig 1A). Furthermore, patients were separated into two groups by an approximate cut-off value of 40% CD4 $T_{HD}$ cells (Fig 1A); we thus denominated "G1 cohort" to patients with more than 40% $T_{HD}$ cells (63.25 ± 13.5%, $N = 23$) and "G2 cohort" to patients with less than 40% (27.05 ± 10.6%, $N = 28$). Differences between G1 and G2 cohorts were also highly significant (Fig 1A).

Objective responders were found only within the G1 cohort ($P = 0.0001$), which included all patients that showed significant tumor regression (Fig 1A and B). Accordingly, ROC analysis demonstrated a highly significant association of the CD4 $T_{HD}$ cell baseline percentage with objective responses ($P = 0.0003$) and confirmed the cut-off value of > 40% to identify objective responders with 100% specificity and 70% sensitivity (Fig 1C).

A validation dataset from 32 patients was performed by parallel independent double-blind sample handling, staining, data collection, and analyses (Fig EV1). While in the discovery cohort T cells were directly analyzed from peripheral blood samples within the same day, validation samples were processed very differently. Briefly, an overnight depletion step of myeloid cells by adherence to plastic was included before T-cell analyses from non-adherent cells. Hence,

relative percentages of CD4 $T_{HD}$ cells varied between the discovery and validation cohorts. Even so, there was a significant agreement between the two datasets on patient classification as demonstrated by Cohen's kappa coefficient ($\kappa = 0.932$). The highly significant association between G1 patients and objective responses in the validation set was confirmed ($P = 0.0006$), albeit with a cut-off value of 20% in the validation dataset which was corroborated by ROC analysis (Fig EV1).

In agreement with these results, the G1 patient cohort had a significantly longer progression-free survival (PFS) compared to the G2 cohort. The median PFS (mPFS) of G2 patients was only 6.1 weeks (95% C.I., 5.7–6.6) compared to 23.7 weeks for G1 patients (95% C.I., 0–51.7; $P = 0.001$; Fig 1D). A comparison of G2 versus G1 baseline profiles showed hazard ratios for disease progression or death that favored the latter [3.1 (1.5–6.4; 95% C.I.) $P = 0.002$].

To assess whether CD4 T-cell profiling had prognostic value, the time elapsed from diagnosis to the start of immunotherapies was compared between G1 and G2 patient cohorts, as described (Le *et al*, 2015). No significant differences were observed, indicating that G1/G2 classification did not have prognostic value (Fig EV2). This was supported by no association between G1/G2 patient cohorts and baseline ECOG score ($P = 0.6$), with liver metastases ($P = 0.88$), with tumor load ($P = 0.19$), or with the Gustave-Roussy immune score (GRIm; $P = 0.14$, Table EV2; Bigot *et al*, 2017). The hazard ratio for progression or death of G2 patients maintained its statistical significance by multivariate analyses (HR 9.739; 95% CI 2.501–37.929) when adjusted for tumor histology, age, gender, smoking habit, liver metastases, number of organs affected, PD-L1 tumor expression, NLR, serum LDH, and albumin.

## Functionality of systemic CD4 immunity defines clinical outcomes and susceptibility to PD-L1/PD-1 blockade

We hypothesized that the relative percentage of CD4 $T_{HD}$ cells was a biomarker for functional differences in systemic CD4 immunity between the two cohorts before the start of immunotherapy. To find out whether this was the case, we first evaluated PD-1 expression in unstimulated CD4 T cells. However, no differences were observed between G1 and G2 patient cohorts or even with healthy age-matched donors (not shown). We then tested whether there were differences in PD-1 upregulation after *ex vivo* stimulation with lung cancer cells. To this end, we engineered a T-cell stimulator cell line by expressing a membrane-bound anti-CD3 single-chain antibody in A549 human lung adenocarcinoma cells (A549-SC3 cells). This cell line stimulated T cells in co-cultures with the same affinity and specificity while preserving other inhibitory interactions such as PD-L1/PD-1 or MHC II-LAG-3 (Fig EV3A and B). This ensured the same standard assay for cancer cell T-cell recognition for each patient (Fig EV3B–D). CD4 T cells from NSCLC patients significantly upregulated PD-1 compared to cells from age-matched healthy donors after incubation with A549-SC3 cells ($P < 0.001$; Figs EV3C and 2A). However, no differences were found between G1 and G2 patient cohorts. Coexpression of PD-1 and LAG-3 has been suggested to identify dysfunctional tumor-infiltrating lymphocytes in NSCLC (He *et al*, 2017). Interestingly, G2 donors presented a significantly higher percentage of CD4 T cells co-expressing both markers than G1 donors after stimulation (Fig 2B). To test whether there were

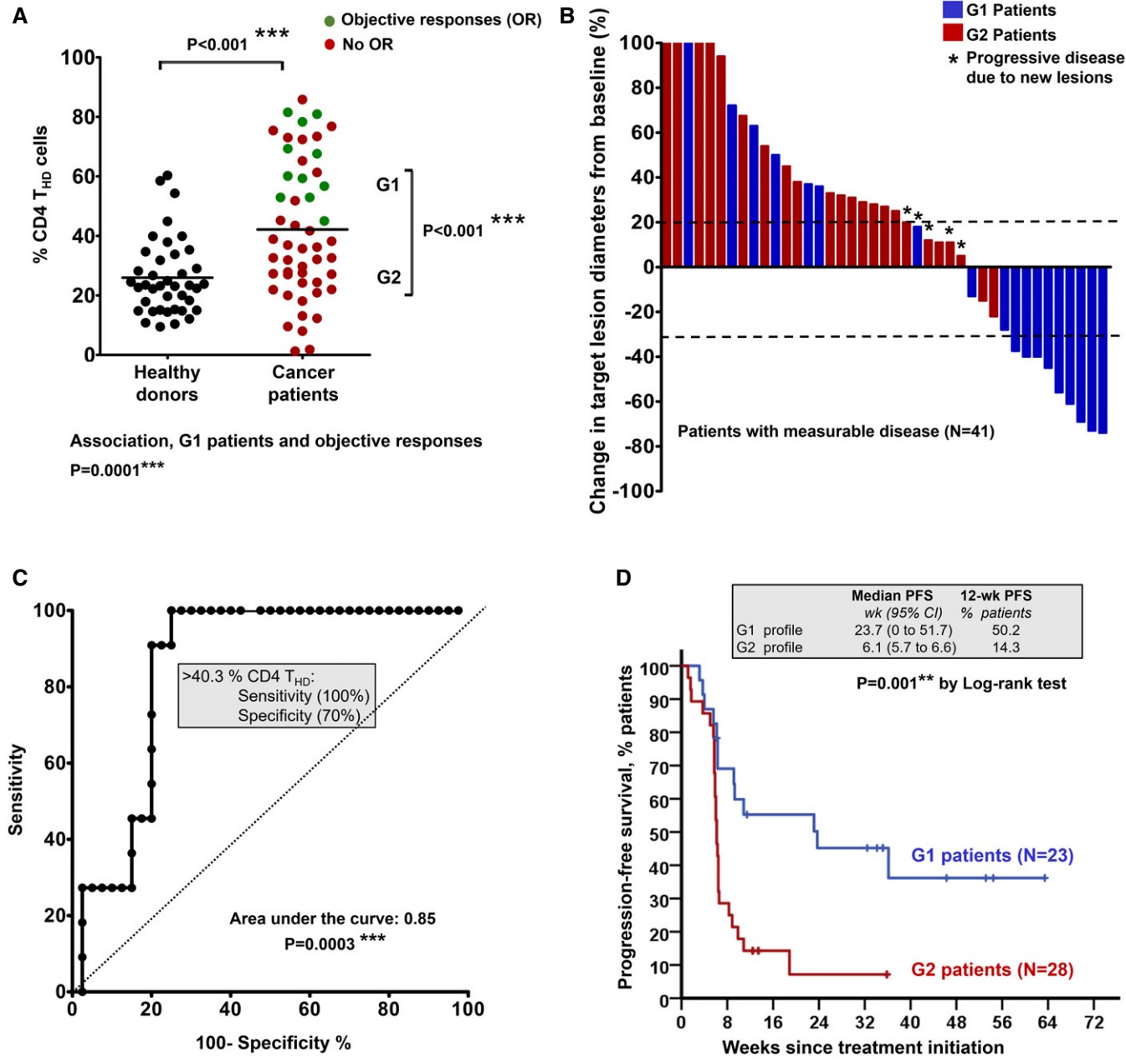

**Figure 1. Baseline profiling of CD4 T-cell differentiation subsets stratifies clinical responses to PD-L1/PD-1 blockade.**

A  Percentage of circulating highly differentiated CD4 T cells within CD4 cells in age-matched healthy donors ($N = 40$) or NSCLC patients ($N = 51$) or NSCLC patients before undergoing immunotherapies. G1 and G2, groups of patients classified according to high $T_{HD}$ cells (G1, > 40% CD4 $T_{HD}$ cells) and low $T_{HD}$ cells (G2, < 40% CD4 $T_{HD}$ cells). Relevant statistical comparisons are shown by the test of Mann–Whitney. In green, objective responders (OR). In red, no OR. Below the graph, correlation of objective responses to G1 and G2 groups by Fisher's exact test.

B  Waterfall plot of change in lesion size in patients with measurable disease classified as having a G1 (blue) or G2 (red) profile. Dotted lines represent the limit to define significant progression (upper line) or significant regression (lower line).

C  ROC analysis of baseline CD4 $T_{HD}$ quantification as a function of objective clinical responses.

D  Kaplan–Meier plot for PFS in patients treated with immunotherapies stratified only by G1 (green) and G2 (red) CD4 T-cell profiles. Patients starting therapy with a G2 profile had an overall response rate (ORR) of 0 and 82% of them experienced progression or death by week 9. ORR was 44.8% for G1 patients, and the 12-week PFS was 50.2%.

Source data are available online for this figure.

also differences in proliferation, the percentage of Ki67$^+$ cells was compared (Fig 2C and D). Accordingly, CD4 T cells from G2 patients were remarkably impaired in proliferation after *ex vivo* activation with A549-SC3 cells compared to T cells from G1 patients. As we had observed that G1 and G2 patient cohorts differed in baseline percentages of CD4 $T_{HD}$ cells (Fig 1A), we tested whether this subset

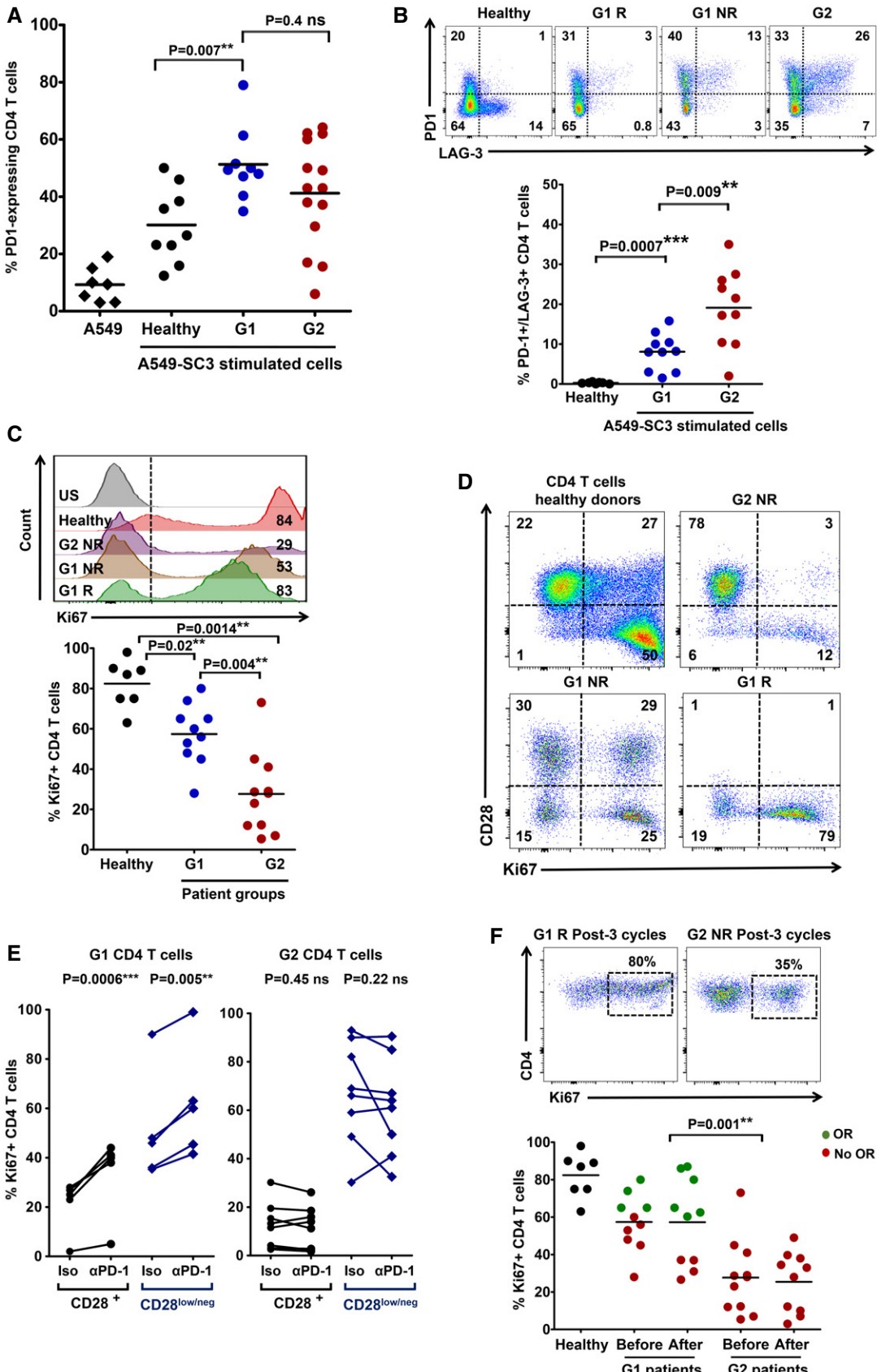

Figure 2.

Figure 2.  Differential systemic CD4 immunity and responses to PD-1/PD-L1 blockade in NSCLC patients.

A   The scatter plot shows PD-1 expression after co-culture of CD4 T cells from healthy donors (n = 9) or NSCLC patients (n = 14), as indicated, with A459-SC3 lung cancer cells. Relevant statistical comparisons with the test of Mann–Whitney are indicated.

B   Upper graphs, flow cytometry density plots of PD-1 and LAG-3 co-expression in CD4 T cells from healthy donors, a G1 responder (G1 R), a G1 non-responder (G1 NR), and a G2 non-responder as indicated, following stimulation with A549-SC3 cells. Percentage of expressing cells are indicated within each quadrant. Below, same as in the upper graphs but as a scatter plot of the percentage of CD4 T cells that simultaneously co-express PD-1 and LAG-3 that simultaneously co-express PD-1 and LAG-3 in G1 healthy donors (n = 10), G1 (n = 10) and G2 (n = 10) patients. Relevant statistical comparisons are shown with the test of Mann–Whitney.

C   Upper flow cytometry histograms of Ki67 expression in CD4 T cells from the representative subjects as indicated on the right, after stimulation with A549-SC3 cells. Vertical dotted line indicates the cut-off value of positive versus negative Ki67 expression. The percentage of Ki67-expressing CD4 T cells is shown within the histograms. Below, same data represented as a scatter plot from a sample of G1 and G2 donors as indicated, with relevant statistical comparisons with the test of Mann–Whitney (n = 7–10).

D   Proliferation of CD4 T cells stimulated by A549-SC3 cells from the indicated patient groups. CD28 expression is shown together with the proliferation marker Ki67. Percentages of cells within each quadrant are shown.

E   Same as in (D) but in the presence of an isotype control antibody or an anti-PD-1 antibody with the equivalent sequence to pembrolizumab. The effects on CD4 T cells from a G1 and a G2 patient are shown, divided into CD28 high or low/negative subsets as indicated. Relevant statistical comparisons are shown with paired Student's t-test.

F   Top, flow cytometry density plots of Ki67 expression in CD4 T cells from representative G1 or G2 patients after three cycles of therapy, activated by incubation with A549-SC3 cells. Below, same as above but as a dot-plot graph (n = 7–10). A comparison between proliferating CD4 T cells before and after therapy is shown in unpaired patient samples. G1 R, G1 objective responder patient. G2 NR, G2 patient with no objective responses; green, objective responders (OR) and red, no OR; Iso, treatment with an isotype antibody control; and α-PD-1, treatment with anti-PD-1 antibody. Statistical comparisons were performed with the test of Mann–Whitney.

was responsive to activation by A549-SC3 cells (Fig 2D). Interestingly, CD4 $T_{HD}$ cells strongly proliferated in all patients, although they constituted a minority in the G2 patient cohort.

The strong proliferative capacities of CD4 $T_{HD}$ cells indicated that these were not exhausted, anergic, or senescent subsets, but probably highly differentiated memory subsets. To test this, their baseline phenotype according to CD62L/CD45RA surface expression was assessed in a sample of patients (Fig EV4A). The majority of CD4 $T_{HD}$ cells were central-memory (CD45RA$^{negative}$ CD62L$^{+}$) and effector-memory (CD45RA$^{negative}$ CD62L$^{negative}$) cells, without differences between G1 and G2 cohorts. Increased genotoxic damage is strongly associated with T-cell senescence and can be evaluated by H2AX expression (Lanna et al, 2017). Interestingly, NSCLC CD4 T cells exhibited extensive genotoxic damage in both $T_{HD}$ and non-$T_{HD}$ subsets without differences between G1 and G2 patient cohorts, unlike T cells from age-matched healthy donors (Fig EV4B). Therefore, genotoxic damage did not identify senescent T cells in patients that had been treated with conventional therapies. Then, the expression of the replicative senescence marker CD57 was used to identify bona fide senescent T cells, which accounted to 30% of $T_{HD}$ cells in healthy age-matched donors, and about 10% in NSCLC patients (Fig EV4C). Our results strongly suggested that circulating CD4 $T_{HD}$ cells in our cohort of NSCLC patients mostly corresponded to non-senescent, non-exhausted memory subsets.

CD4 T cells of G2 patients strongly co-upregulated PD-1/LAG-3 after stimulation. We wondered if lack of clinical responses in G2 patients could be explained by resistance to single blockade of PD-1. Hence, proliferation of CD4 T cells activated with A549-SC3 in the presence of an anti-PD-1 antibody equivalent to pembrolizumab was assessed (Scapin et al, 2015; Fig 2E). As expected, PD-1 blockade increased proliferation of $T_{HD}$ and non-$T_{HD}$ CD4 T cells in patients from the G1 cohort. In contrast, their G2 counterparts were largely refractory. To find out whether CD4 T cells from G2 patients remained unresponsive to PD-1 blockade in vivo, cells were obtained from patients after at least three cycles of therapy and tested for their proliferative capacities (Fig 2F). Systemic CD4 T cells from G2 patients remained poorly proliferative during immunotherapy.

## Absence of cancer-specific CD4 T cells or systemic T-cell exhaustion is not behind the lack of objective clinical responses to PD-L1/PD-1 blockade therapies

Then, we thought that G2 patients could be refractory to anti-PD-1 immunotherapy by not having systemic cancer-specific CD4 T cells. To this end, we quantified CD4 T cells reactive to lung adenocarcinoma antigens using IFN-γ-activated autologous monocyte-derived DCs as antigen presenting cells, as described (Escors et al, 2008). DCs were loaded with A549 cell lysate, as these cells contain numerous common lung adenocarcinoma antigens (Madoz-Gurpide et al, 2008). We used this approach as we lacked sufficient biopsy material to get tumor antigens or tumor-infiltrating T cells. CD4 T cells reactive to A549 cell antigens were identified by IFN-γ upregulation. Interestingly, lung cancer-specific CD4 T cells were present at varying proportions before the start of immunotherapy in both G1 and G2 patients (Fig 3A). Indeed, although the average percentages of circulating lung cancer-specific CD4 T cells were low, these did not differ significantly between G1 (responders and non-responders) and G2 patients. These T cells consisted of both $T_{HD}$ and non-$T_{HD}$ subsets, without significant differences in relative percentages between G1 and G2 cohorts (Fig 3B). These results suggested that poor responses in G2 patients were not caused by lack of tumor-specific CD4 T cells but rather by having dysfunctional T cells.

To further study the dysfunctional status of systemic CD4 T cells in G2 patients, we evaluated PD-1 and LAG-3 surface expression directly after blood sampling, as constitutive high-level expression of these markers is a frequent characteristic of T-cell exhaustion. However, no differences were found between age-matched healthy donors and G1/G2 patient cohorts in either $T_{HD}$ or non-$T_{HD}$ subsets (not shown). Nevertheless, the defining hallmark of T-cell exhaustion is the loss of cytokine production following stimulation, particularly multi-cytokine expression (Crawford et al, 2014). Interestingly, CD4 T cells from both G1 and G2 patient cohorts were as proficient in IFN-γ, IL-4, IL-10, and IL-2 expression as T cells from healthy donors independently of their CD28 expression (Fig 4A) or whether these were T cells from G1 responders or non-responders (Appendix Fig S1). Indeed, CD4 cells (total, $T_{HD}$, and non-$T_{HD}$ subsets) in both G1 and G2 patient cohorts were significantly skewed

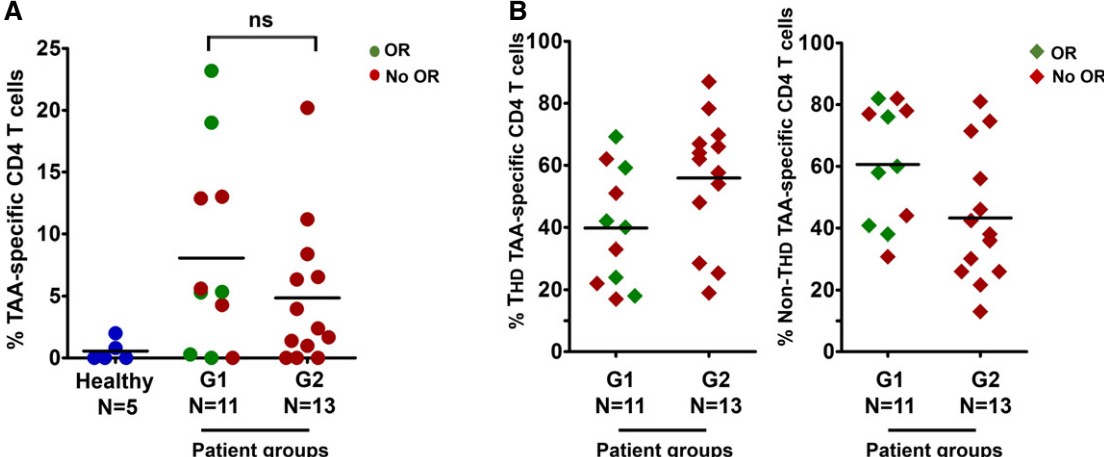

**Figure 3. Lung cancer antigen-specific CD4 T cells in NSCLC patients.**

A   Scatter plot graph with the percentage of lung cancer-specific systemic CD4 T cells quantified by an autologous DC-based antigen presentation assay (see Materials and Methods), in a sample of G1 and G2 patients as indicated. Objective responses (OR) are shown in green. In red, patients with no OR.

B   The scatter plot graph on the left represents the percentage of CD4 $T_{HD}$ cells within lung cancer-specific CD4 T cells in a sample of patients from the indicated G1/G2 groups. On the right, same as left but representing the percentage of CD28$^+$ CD4 T cells within lung cancer-specific CD4 T cells. Objective responders (OR) are shown in green. In red, patients with no OR.

Data information: Relevant statistical comparisons are shown within the graphs with the test of Mann–Whitney. *N*, number of biological replicates (independent patients); Ns, no significant differences (*P* < 0.05).

toward IL-17 responses compared to age-matched healthy donors (Fig 4A). Importantly, only a minority of CD4 T cells from either G1 or G2 patient groups were single-cytokine producers (Fig 4B) while most of the non-$T_{HD}$ CD4 T cells were very proficient in multiple cytokine production with a preference for IL-17-expressing subsets (Fig 4C and D). These results indicated that CD4 T cells from G2 patients were not exhausted according to our current understanding (Hashimoto *et al*, 2018). Indeed, they responded to stimulation by producing cytokines although with strong co-upregulation of PD-1/LAG-3 associated with markedly diminished proliferative capacities.

**Systemic CD8 immunity recovers in G1 responder patients following immunotherapy**

In contrast to CD4 $T_{HD}$ cells, the relative percentage of CD8 $T_{HD}$ cells within the CD8 population did not significantly differ from age-matched healthy donors, nor could be used to identify objective responders (Fig EV5A and B). Interestingly, CD8 cells from both G1 and G2 patient cohorts obtained before the start of immunotherapies did fail to proliferate after stimulation by A549-SC3 cells (Fig 5A). To test whether anti-PD-1 therapy could recover CD8 dysfunctionality *in vivo*, the proliferative capacities of CD8 T cells from G1 and G2 patients obtained after at least three cycles of treatment were evaluated by stimulation with A549-SC3 cells. CD8 T cells from G1 responders had recovered significant proliferative capacities, while only limited enhancements were observed in G2 patients (Fig 5B). Similarly to CD4 cells, systemic CD8 T cells specific for lung adeno-carcinoma antigens were quantified in G1 and G2 patients and found to be comparable (Fig EV5C) and distributed within non-$T_{HD}$ and $T_{HD}$ subsets (Fig EV5D).

To find out whether CD8 T cells in G1 patients were especially susceptible to PD-1 blockade *ex vivo*, baseline samples of CD8 T cells

from G1 and G2 patients were activated with A549-SC3 cells in the presence of an anti-PD-1 antibody or an isotype control. In agreement with the *in vivo* results, *ex vivo* PD-1 blockade improved significantly the proliferation of CD8 T cells from G1 patients and specially non-$T_{HD}$ (CD28$^+$) subsets (Fig 5C). *In vivo* expansion of CD28$^+$ CD8 T cells in murine models correlate with anti-PD-1 efficacy (Kamphorst *et al*, 2017b). To confirm this observation in our cohort of patients, the changes in the relative abundance of CD8 CD28$^+$ T cells were compared in G1 and G2 patients from baseline to post-anti-PD-1 therapy (Fig 5D). Accordingly, the CD28$^+$ CD8 T-cell compartment significantly expanded (*P* < 0.001) only in G1 patients.

**Proliferative dysfunctionality of CD4 and CD8 T cells from G2 patients is reversible after PD-1/LAG-3 dual blockade**

As we found that CD4 proliferative dysfunctionality in G2 patients correlated with high PD-1/LAG-3 co-upregulation after activation, we tested if this was also the case for CD8 T cells. PD-1/LAG-3 co-expression was tested *ex vivo* after stimulation with A549-SC3 cells, and G2 patients presented a significantly higher proportion of PD-1/LAG-3 co-expressing CD8 T cells compared to G1 counterparts (Fig 6A). Overall, our data indicated that PD-1/LAG-3 co-upregulation was contributing to proliferative dysfunctionality. To test whether this was the case, baseline samples of CD4 and CD8 T cells from G2 patients were co-incubated *ex vivo* with A549-SC3 cells in the presence of an isotype antibody control, anti-PD-1, anti-LAG-3, or anti-PD-1/anti-LAG-3 antibodies. We confirmed that each antibody was specifically blocking PD-1, LAG-3, or both in our assays by epitope masking using flow cytometry (not shown). Only co-blockade of PD-1 and LAG-3 in both CD4 (Fig 6B) and CD8 T cells (Fig 6C) from G2 patients significantly increased proliferation independently of CD28 expression. These results

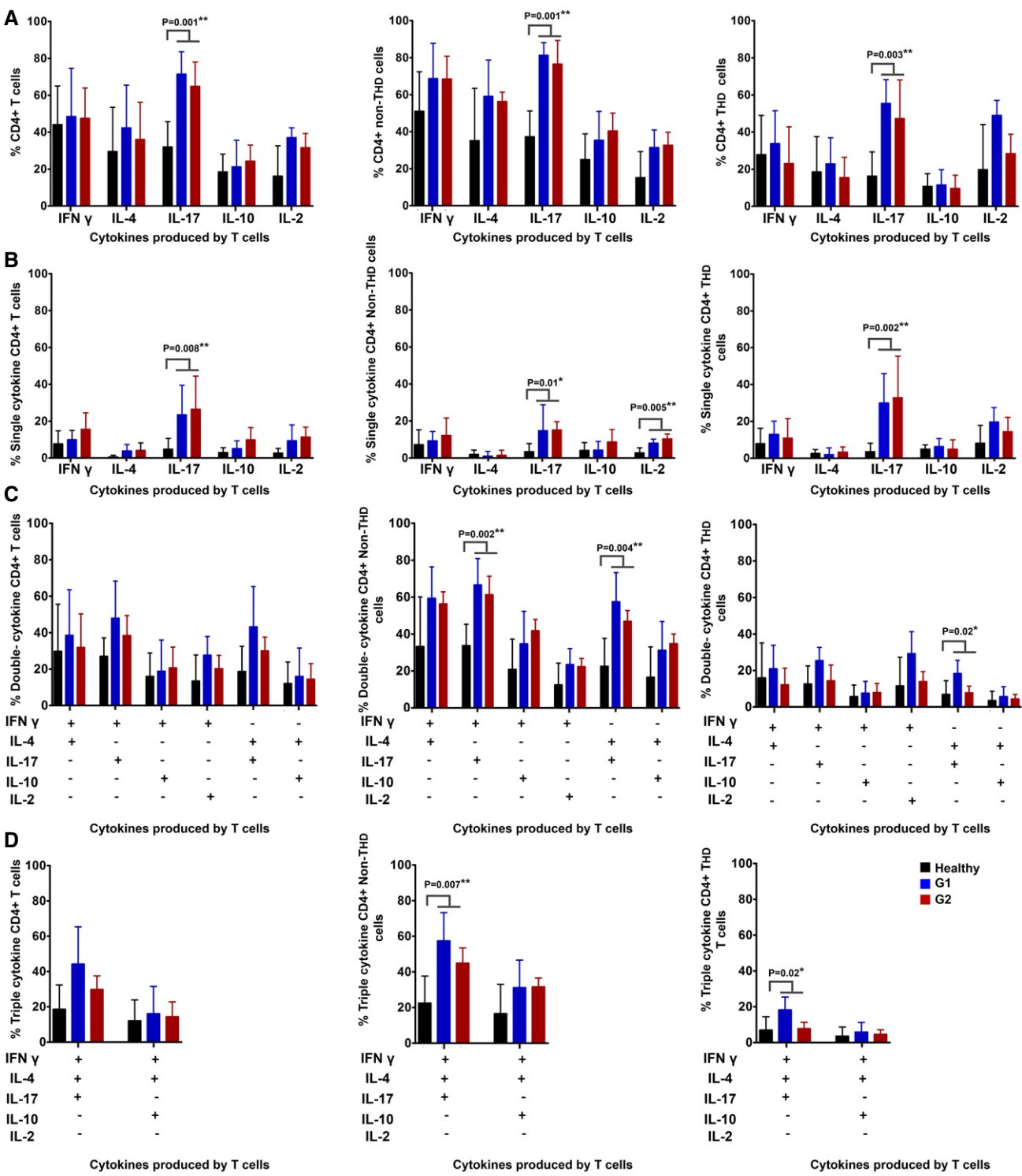

**Figure 4. Systemic circulating CD4 T cells in NSCLC patients are proficient in cytokine production with an overall Th17 profile.**

A    Column graphs representing the percentage of CD4 T cells from NSCLC patients or age-matched healthy donors as represented in the graph, expressing the indicated cytokines after T-cell stimulation with anti-CD3/anti-CD28 antibodies. Data on total CD4 (left graph), CD28+ subsets (center graph) and CD28negative subsets (right graph) are shown. Error bars correspond to standard deviations, and bars represent means from nine independent biological replicates (healthy donors) and six independent replicates (patients).

B–D    Same as in (A) but representing CD4 T cells expressing only one cytokine (B), two (C) or three cytokines simultaneously (D). Error bars correspond to standard deviations, and bars represent means from five independent biological replicates (patients).

Data information: Relevant statistical comparisons are shown within the graphs by the test of Kruskal–Wallis.

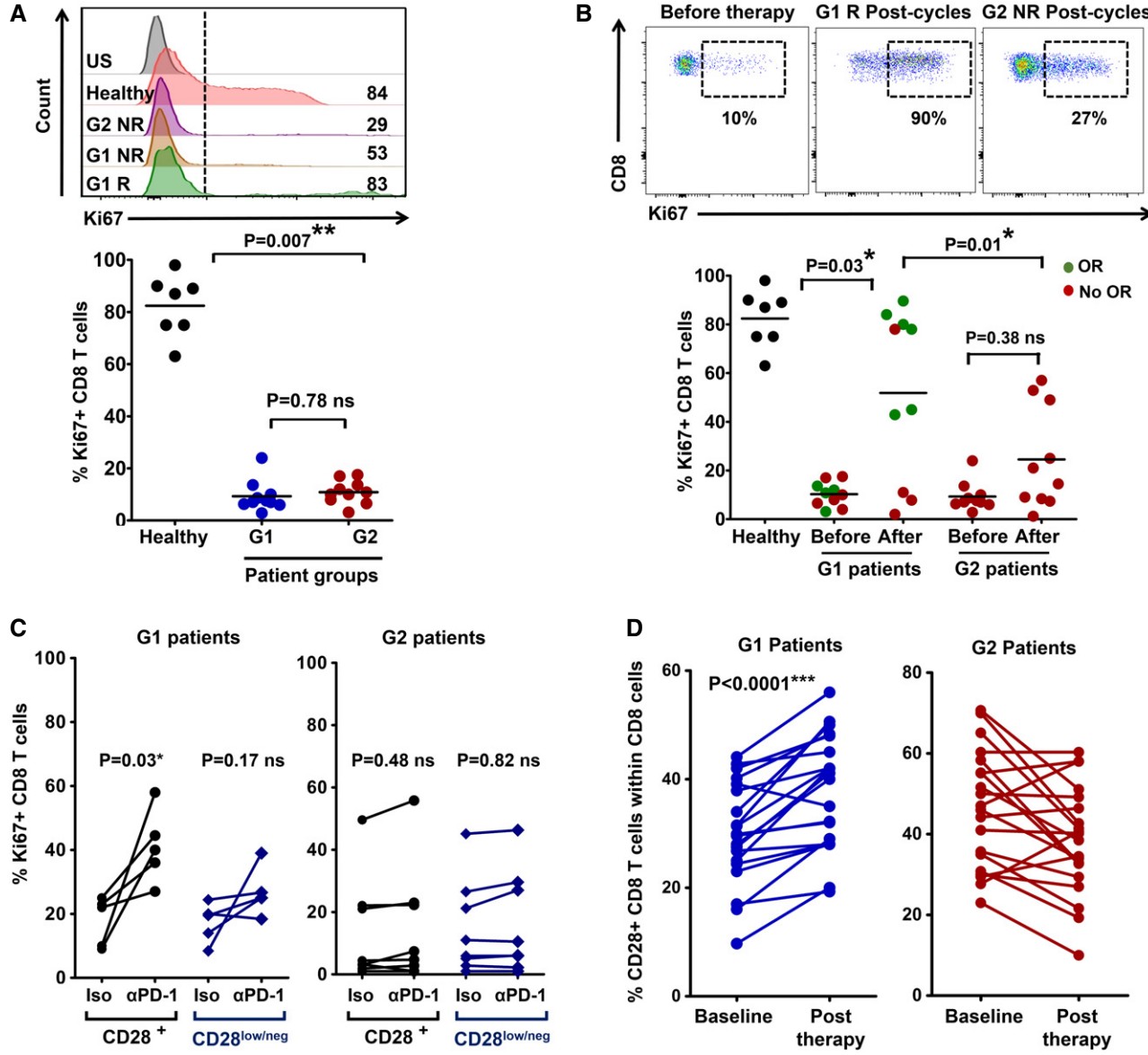

**Figure 5. CD8 dysfunctionality recovers in G1 patients undergoing immunotherapy.**

A  Upper flow cytometry histograms, expression of the proliferation marker Ki67 in CD8 T cells from the indicated patients or healthy donor before the start of immunotherapy, stimulated *ex vivo* by A549-SC3 cells. Numbers indicate mean fluorescence intensities. G1 R and G1 NR, responder and non-responder G1 patient, respectively; G2 NR, non-responder G2 patient. US, unstained control. Below, same as above but as a dot plot graph with percentage of proliferating Ki67+ CD8 T cells from the indicated groups (*n* = 7–10). Relevant statistical comparisons are shown with the test of Mann–Whitney.

B  Upper flow cytometry density plots, expression of Ki67 in *ex vivo*-stimulated CD8 T cells from the indicated patients before and after the start of immunotherapies. NR, non-responder patient; R, responder patient. Below, dot-plots of the percentage of Ki67+ proliferating CD8 T cells after *ex vivo* activation by A549-SC3 cells. CD8 T cells were obtained from samples of G1 or G2 patients before immunotherapy and after three cycles of anti-PD-1 therapy (*n* = 7–10). Relevant statistical comparisons are shown with the test of Mann–Whitney. Green, objective responders (OR) and red, no ORs.

C  Same as in (A) but in the presence of an isotype control antibody or an anti-PD-1 antibody molecularly equivalent to pembrolizumab. Relevant statistical comparisons are shown with comparisons carried out with paired Student's *t*-test.

D  Change in CD8 CD28+ T cells from baseline to post-therapy in G1 patients (left) or in G2 patients (right). Statistical comparisons were carried out with paired Student's *t*-test.

confirmed that PD-1/LAG-3 co-upregulation contributed to keeping systemic CD4 and CD8 T cells from G2 patients in a proliferative dysfunctional state following stimulation, and that this T-cell dysfunctionality can be reverted by co-blockade of both immune checkpoints.

**Objective responders are found within G1 patients with PD-L1-positive tumors**

Objective response rates in G1 patients were about 50%. Hence, our results indicated that functional systemic CD4 responses

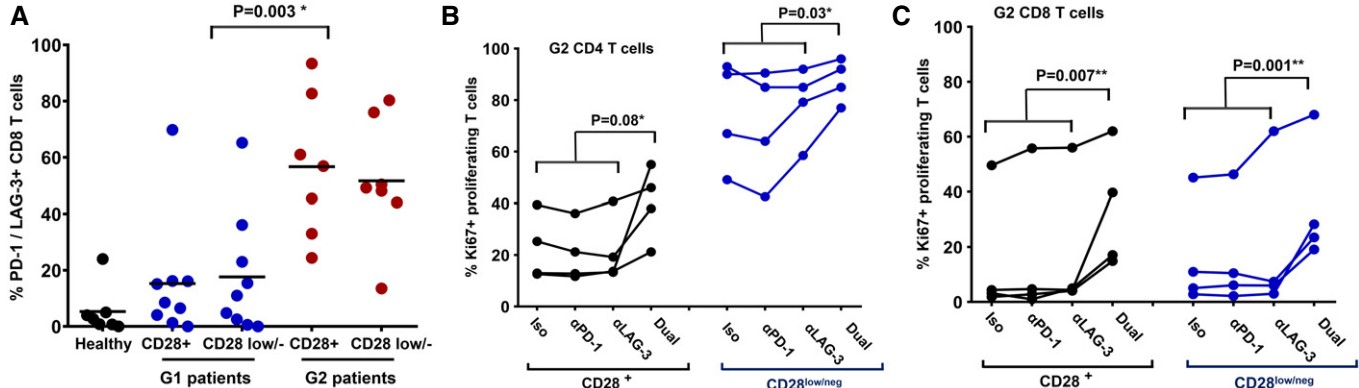

**Figure 6.  PD-1/LAG-3 co-blockade recovers proliferative capacities of CD4 and CD8 T cells from G2 patients.**

A    Scatter plots of PD-1/LAG-3-expressing CD8 T cells after activation by A459-SC3 cells in a sample of G1 (n = 9) and G2 (n = 7) patients within CD28[+] and CD28[negative] populations as indicated in the figure. Relevant statistical comparisons are shown with the test of Kruskal–Wallis.

B, C  Dot-plot representing the percentage of proliferating CD4 T cells (B) and CD8 T cells (C) from a sample of G2 patients before starting immunotherapy, activated ex vivo by A549-SC3 cells in the presence of the indicated antibodies. "Dual" represents the addition of both anti-PD-1 and anti-LAG-3 antibodies. Appropriate statistical comparisons are shown within the graph with two-way paired ANOVA. Data from CD28[+] and CD28[negative] subsets are represented as indicated.

were necessary but not sufficient for clinical efficacy. As NSCLC patients with high PD-L1 tumor expression benefit from anti-PD-L1/PD-1 blockade therapies (Borghaei et al, 2015), we assessed PD-L1 tumor expression and its association to responses in G1 and G2 patient cohorts for whom PD-L1 tumor expression could be determined. G1 patients with PD-L1-positive tumors had a PFS of 70% (> 5%; P = 0.007; Fig 7A). The same benefit was observed when the stratification was extended to include patients with unknown PD-L1 tumor status in our cohort (Fig 7B).

## Discussion

Tumor intrinsic and extrinsic factors contribute to the efficacy of PD-L1/PD-1 blockade therapies. So far, not a single factor has been

associated with objective responses or progression, suggesting that multiple mechanisms influence clinical responses.

Because PD-L1/PD-1 blocking antibodies are systemically administered, these therapies cause systemic changes in immune cell populations (Kamphorst et al, 2017a; Krieg et al, 2018). Some of these changes may reflect the efficacy of immunotherapy in patients and could be used for patient stratification. Several studies have been performed to monitor systemic dynamics of immune cell populations, some of them retrospectively and by high-throughput techniques (Hui et al, 2017; Kamphorst et al, 2017b; Krieg et al, 2018). We evaluated responses from fresh blood samples because freezing PBMCs led to a significant alteration in the distribution of immune cell types, and distorted expression patterns of cell surface markers. Hence, sample manipulation had a significant impact on our results, which limited our study to prospective data.

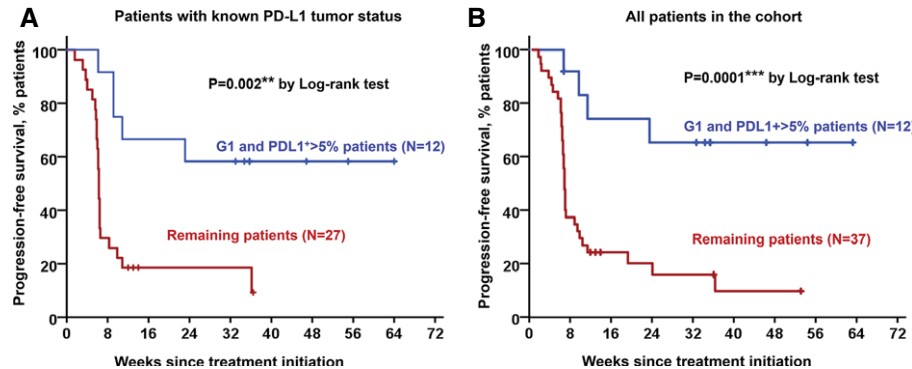

**Figure 7.  Objective responders are found within G1 patients with PD-L1[+] tumors.**

A    Kaplan–Meier plot for PFS in patients undergoing immune checkpoint inhibitor therapies stratified by G1/PD-L1[+] tumors (blue) and remaining patients for whom their PD-L1 tumor status is known (red).

B    Same as in (A) but including all patients in the study cohort. Remaining patients (red) also included G1 patients with PD-L1 low/negative tumors, G1 patients with unknown PD-L1 tumor status, and G2 patients with either PD-L1[+] or PD-L1-negative tumors.

Source data are available online for this figure.

Here, we found that the proliferative functionality of systemic CD4 immunity is required for clinical responses to PD-L1/PD-1 blockade therapy. Indeed, it was a differential baseline factor in our cohort of NSCLC patients progressing from conventional therapies. Hence, patients with non-dysfunctional CD4 responses contained all objective responders with a response rate of about 50% (G1 patients), while no objective responses were observed in patients with dysfunctional CD4 T cells (G2 patients). CD4 T-cell dysfunctionality in G2 patients was reflected as strongly impaired proliferation after stimulation, high co-expression of LAG-3/PD-1, and resistance to *ex vivo* and *in vivo* PD-1 monoblockades. As both responders and non-responders contained comparable proportions of lung cancer-specific CD4 and CD8 T cells in our cohort of patients before the start of therapy, the experimental evidence pointed to the baseline intrinsic functionality of CD4 immunity as the key factor in our study. Systemic CD28$^+$ CD4 T cells in G2 patients were not truly exhausted or *bona fide* anergic T cells. No constitutive high-level expression of PD-1 and LAG-3 was observed unless stimulated. They were proficient in multi-cytokine expression following stimulation. Indeed, CD4 T cells from both G1 and G2 patient cohorts were skewed toward Th17-expressing phenotypes compared to healthy donors. All these characteristics were indicators of systemic CD4 T-cell proliferative dysfunctionality in G2 patients.

Importantly, patients with functional CD4 immunity could be easily identified by having a high proportion of circulating CD4 T$_{HD}$ memory cells. ROC analysis provided a cut-off value of > 40% CD4 T$_{HD}$ to identify objective responders from freshly analyzed blood samples. It is worth noting that the cut-off value was reduced to 20% in a validation cohort that was independently processed and analyzed by a very different procedure. Importantly, patient classification in G1 or G2 cohorts and their association with clinical responses agreed independently of the protocol utilized. We are well aware that quantification of CD4 T$_{HD}$ cells could be used as a baseline factor for clinical stratification. Proper validation of CD4 T-cell profiling will require protocol standardization for sample manipulation and analyses. In fact, G1 patients with PD-L1-positive tumors exhibited response rates of 70%, which strongly highlights the role of CD4 immunity in clinical responses. However, the main goal of the current study was to understand the contribution of systemic T-cell immunity to PD-L1/PD-1 blockade therapies, rather than providing a predictive biomarker.

The requirement of functional systemic immunity has been previously demonstrated in murine models for the efficacy of other immunotherapy approaches (Spitzer *et al*, 2017), as well as the importance of CD4 T cells for anti-PD-1 immunotherapy (Markowitz *et al*, 2018). These studies are in agreement with our present data in human patients undergoing PD-L1/PD-1 blockade therapies. Indeed, the appearance of a specific murine subtype of CD4 T cell was the main correlator with efficacious responses by administration of anti-cancer cell immunoglobulins (Spitzer *et al*, 2017). These results together with our data strongly support the need for proficient CD4 responses to achieve efficacious responses.

Immune checkpoint inhibitor therapy aims to recover CD8 cytotoxic responses (Ahmadzadeh *et al*, 2009). To our surprise, all systemic CD8 T cells in patients before the start of immunotherapies were also dysfunctional. Nevertheless, the proliferative capacities of CD8 T cells were recovered during immunotherapy but only in patients with functional CD4 immunity. This was reflected by an expansion of CD28$^+$ cells in agreement with data in murine models (Kamphorst *et al*, 2017b). CD8 dysfunctionality in G2 patients was again correlated with PD-1/LAG-3 co-upregulation. Both CD4 proliferative dysfunctionality and CD8 proliferative dysfunctionality in G2 patients were reversible *ex vivo* by PD-1/LAG-3 co-blockade.

An increasing number of studies are linking PD-1/LAG-3 co-expression in T cells to resistance to anti-PD-L1/PD-1 therapies (Mishra *et al*, 2016; Huang *et al*, 2017; Williams *et al*, 2017; Johnson *et al*, 2018). Our study prompts the clinical evaluation of patients with systemic CD4 T-cell dysfunctionality by PD-1/LAG-3 dual-blockade strategies.

# Materials and Methods

### Study design

The study was approved by the Ethics Committee at the Hospital Complex of Navarre. Informed consent was obtained from all subjects and all experiments conformed to the principles set out in the WMA Declaration of Helsinki and the Department of Health and Human Services Belmont Report. Samples were collected by the Blood and Tissue Bank of Navarre, Health Department of Navarre, Spain. Thirty-nine patients diagnosed with non-squamous and 12 with squamous NSCLC were recruited at the Hospital Complex of Navarre (Table EV1). Patients had all progressed to first-line chemotherapy or concurrent chemo-radiotherapy. Eligible patients were 18 years of age or older who agreed to receive immunotherapy targeting PD-1/PD-L1 following the current indications (Table EV1). Tumor PD-L1 expression could be quantified in 39 of these patients before the start of therapies. Measurable disease was not required. The exclusion criteria consisted of concomitant administration of chemotherapy or previous immunotherapy treatment. NSCLC patients had an age of $65 \pm 8.9$ ($N = 51$). Age-matched healthy donors were recruited from whom written informed consent was also obtained, with an age of $68.60 \pm 8$ (mean $\pm$ SD, $N = 40$).

Therapy with nivolumab, pembrolizumab, and atezolizumab was provided following current indications (Herbst *et al*, 2016; Horn *et al*, 2017; Rittmeyer *et al*, 2017). 4 ml peripheral blood samples were obtained prior and during immunotherapy before administration of each cycle. PBMCs were isolated as described (Escors *et al*, 2008) and T cells analyzed by flow cytometry. The participation of each patient concluded when a radiological test confirmed response or progression, with the withdrawal of consent or after death of the patient. Tumor responses were evaluated according to RECIST 1.1 (Eisenhauer *et al*, 2009) and Immune-Related Response Criteria (Wolchok *et al*, 2009). Objective responses were confirmed by at least one sequential tumor assessment.

### Flow cytometry

Surface and intracellular flow cytometry analyses were performed as described (Karwacz *et al*, 2011; Gato-Canas *et al*, 2017). T cells were immediately isolated and stained. 4 ml blood samples were

collected from each patient, and PBMCs were isolated by FICOL gradients right after the blood extraction. PBMCs were washed and cells immediately stained with the indicated antibodies in a final volume of 50 µl for 10 min in ice. Cells were washed twice, resuspended in 100 µl of PBS, and analyzed immediately. The following fluorochrome-conjugated antibodies were used at 1:50 dilutions unless otherwise stated: CD4-FITC (clone M-T466, reference 130-080-501, Miltenyi Biotec), CD4-APC-Vio770 (clone M-T466, reference 130-100-455, Miltenyi Biotec), CD4-PECy7 (clone SK3, reference 4129769, BD Biosciences), CD14-VF450 (1:500 dilution, clone 61D3, TONBO), CD3-APC (clone REA613, reference 130-113-135, Miltenyi Biotec), CD27-APC (clone M-T271, reference 130-097-922, Miltenyi Biotec), CD27-PE (clone M-T271, reference 130-093-185, Miltenyi Biotec), CD45RA-FITC (reference 130-098-183, Miltenyi Biotec), CD62L-APC (reference 130-099-252, Miltenyi Biotech), CD28-PECy7 (clone CD28.2, reference 302926, BioLegend), PD-1-PE (clone EH12.2H7, reference 339905, BioLegend), CD8-FITC (clone SDK1, reference 344703, BioLegend), CD8-APC-Cy7(clone RFT-8, reference A15448, Molecular probes by Life technologies), CD57-PE (clone HCD57, reference 322311, BioLegend), H2AX-FITC (1:100 dilution, clone 2F3, reference 613403, BioLegend), LAG-3-PE (clone 11C3C65, reference 369306, BioLegend), IL-2 Alexa Fluor 647 (1:100 dilution, clone MQ1-17H12, reference 500315, BioLegend), IFN γ-APC (1:100 dilution, clone 4S.B3, reference 50256, BioLegend), IFN γ-FITC (1:100 dilution, clone 4S.B3, reference 502506, BioLegend), IL-17A-BV421 (1:100 dilution, clone BL168, reference 512322, BioLegend), IL-17A-Violet 667, clone CZ8-23G1, reference 130-120-554, Miltenyi Biotec), IL-4-PE (1:100 dilution, reference 130-091-647, Miltenyi Biotec), and IL-10-APC (1:100 dilution, reference 130-096-042, Miltenyi Biotec).

## Cell culture

Human lung adenocarcinoma A549 cells were a kind gift of Prof Ruben Pio and authenticated by his group, and were grown in standard conditions. They were confirmed to be mycoplasma-free by PCR. These cells were modified with a lentivector encoding a single-chain version of a membrane-bound anti-OKT3 antibody (Arakawa *et al*, 1996). The lentivector expressed the single-chain antibody construct under the control of the SFFV promoter and puromycin resistance from the human ubiquitin promoter in a pDUAL lentivector construct (Karwacz *et al*, 2011). The single-chain antibody construct contained the variable light and heavy OKT3 immunoglobulin sequences separated by a G-S linker fused to a human IgG1 constant region sequence followed by the PD-L1 transmembrane domain.

Monocyte-derived DCs were generated from adherent mononuclear cells in the presence of recombinant GM-CSF and IL-4 as described (Escors *et al*, 2008). DCs were loaded with A549 protein extract obtained after three cycles of freezing/thawing. Loading was carried out overnight, and DCs were matured with 10 ng/ml of IFN-γ before adding T cells in a 1:3 ratio as described (Escors *et al*, 2008).

When indicated, PD-1 (clone EH12.2H7, BioLegend) and LAG-3 (clone 17B4, BioLegend) blocking antibodies were added to cell cultures at a final concentration of 5 µg/ml. When appropriate, T cells were stimulated with plate-bound anti-CD3/anti-CD28 antibodies as described (Liechtenstein *et al*, 2014).

## Anti-PD-1 antibody production and purification

To generate an antibody molecularly equivalent to the published sequence of pembrolizumab, cDNAs encoding the published amino acid sequences of the heavy and light immunoglobulin chains (Scapin *et al*, 2015) were cloned and expressed in Chinese hamster ovary (CHO) cells. Supernatants were collected and antibodies purified by affinity chromatography following standard procedures.

## Data collection and statistics

T-cell percentages were quantified using FlowJo (Lanna *et al*, 2014, 2017). The percentage of CD4/CD8 $T_{HD}$ (CD28 and CD27 double-negative) and poorly differentiated T cells (CD28$^+$ CD27$^+$) were quantified prior to therapy (baseline) and before administration of each cycle of therapy within CD4 and CD8 cells. Gates in flow cytometry density plots were established taking CD27$^+$ CD28$^+$ T cells as a reference. Data were recorded by M.Z. and separately analyzed thrice by M.Z. and H.A. independently. Cohen's kappa coefficient was utilized to test the inter-rater agreement in classification of immunological profiles ($\kappa = 0.939$).

The mode of action, pharmacokinetics, adverse events, and efficacies of the three PD-L1/PD-1 blocking agents are comparable in NSCLC, which act through the interference with the inhibitory interaction between PD-L1 and PD-1 (Herbst *et al*, 2016; Horn *et al*, 2017; Rittmeyer *et al*, 2017). Treatments administered to the patients were allocated strictly on the basis of their current indications and independently of any variable under study. All data were pre-specified to be pooled to enhance statistical power, and thereby reducing type I errors from testing the hypotheses after *ad hoc* subgrouping into specific PD-L1/PD-1 blockers. The number of patients assured statistical power for Fisher's exact test of 0.95 and superior for Student's *t* and Mann–Whitney tests (G*Power calculator; Faul *et al*, 2009), taking into account that the expected proportion of responders is around 25–35% without stratification (Herbst *et al*, 2016; Horn *et al*, 2017; Rittmeyer *et al*, 2017). Two pre-specified subgroup analyses in the study were contemplated: the first, baseline T-cell values and the second, post-first cycle T-cell changes from baseline. The study protocol contemplated the correlation of these values with responses using Fisher's exact test, paired Student's *t*-tests/repeated-measures ANOVA (if normally distributed) or *U* of Mann–Whitney/Kruskal–Wallis (if not normally distributed, or data with intrinsic high variability). Two-tailed tests were applied with the indicated exceptions (see below).

The percentage of T-cell subsets in untreated cancer patients was normally distributed (Kolmogorov–Smirnov normality test), but not in age-matched healthy donors. Hence, to compare T-cell values between two independent cancer patient groups, two-tailed unpaired Student's *t*-tests were used, while comparisons between healthy subjects and cancer patients were carried out with the Mann–Whitney *U*. Percentages of T-cell populations in treated patients were not normally distributed, so response groups were compared with either Mann–Whitney (comparisons between two independent groups) or Kruskal–Wallis for multi-comparison tests if required. Two-tailed paired *t*-tests were carried out to compare changes in the proportion of CD28$^+$ CD8 T cells between baseline and post-therapy paired groups, and to compare Ki67 expression in T-cell subsets activated with A549-SC3 cells subjected to PD-1 or

**The paper explained**

**Problem**
Over 70% of lung cancer patients progressing from conventional therapies do not respond to PD-L1/PD-1 blockade therapies. The reasons behind this failure are currently unclear.

**Results**
We studied systemic CD4 immunity as a differential factor for clinical responses to PD-L1/PD-1 blockade therapies. Patients with high percentages of systemic highly differentiated memory CD4 T cells contained all responders. In contrast, patients with low percentages of this subset did not respond to therapy. These patients were refractory to immunotherapy, and their systemic CD4 T cells failed to proliferate following activation. Furthermore, they responded by strongly co-upregulating PD-1 and LAG-3. We demonstrated that these T cells could overcome their proliferative dysfunctionality by PD-1/LAG-3 co-blockade with antibodies.

**Impact**
Profiling of CD4 T-cell subsets can help identifying patients with a high probability of responding to immunotherapy, especially in combination with PD-L1 tumor expression. Patients with dysfunctional CD4 immunity before starting PD-L1/PD-1 blockade could undergo alternative therapies such as combinations with LAG-3 blocking agents.

LAG-3 blockade. For comparison of paired samples with anti-PD-1/anti-LAG-3 combinations, two-way ANOVA tests with a random criterium (subjects) were used. Fisher's exact test was used to assess the association of the baseline values of $T_{HD}$ cells with clinical responses. The same tests were performed to assess associations between G1/G2 groups with the indicated prognostic variables.

Progression-free survival (PFS) was defined as the time from the starting date of therapy to the date of disease progression or the date of death by any cause, whichever occurred first. PFS was censored on the date of the last tumor assessment demonstrating absence of progressive disease in progression-free and alive patients. PFS rates at 12 and 28 weeks were estimated as the proportion of patients who were free-of-disease progression and alive at 12 and 28 weeks after the initiation of immunotherapies. Patients who dropped out for worsening of disease and did not have a 28-week tumor assessment were considered as having progressive disease. Overall response rate (ORR) was the proportion of patients who achieved best overall response of complete or partial responses.

PFS was represented by Kaplan–Meier plots and log-rank tests utilized to compare cohorts. Hazard ratios were estimated by Cox regression models. Receiver operating characteristic (ROC) analysis was performed with baseline $T_{HD}$ numbers and response/no response as a binary output. Statistical tests were performed with GraphPad Prism 5 and SPSS statistical packages.

### Validation dataset

Data from a set of 32 patients were validated in parallel by independent handling, processing, staining, flow cytometry data collection, and analysis. The validation dataset was generated by a technician working in unrelated research themes (A.B.). A very different protocol was used to quantify CD4 $T_{HD}$ cells in the validation set compared to the discovery cohort. For the validation dataset, isolated PBMCs were resuspended in TeXmacs serum-free medium (Miltenyi) and plated on 6-well cell culture plates. Myeloid cells were allowed to adhere overnight, and non-adherent cells were collected, centrifuged, and resuspended and T cells stained with the appropriate antibodies for flow cytometry analyses. ROC analysis was used to establish the cut-off value for the relative percentage of CD4 $T_{HD}$ cells to discriminate G1 versus G2 patients in the validation cohort. *Post hoc* Cohen's kappa coefficient test was used to test the agreement between the discovery cohort versus the validation cohort on classification of G1/G2 patients.

**Expanded View** for this article is available online.

### Acknowledgements

We sincerely thank the patients and families that generously agreed to take part in this study. We also thank the Blood and Tissue Bank of Navarre, Health Department of Navarre, Spain. We are thankful to Drs Luis Montuenga and Ruben Pio for their constructive comments and input. This research was supported by Asociación Española Contra el Cáncer (AECC, PROYE16001ESCO); Instituto de Salud Carlos III, Spain (FIS project grant PI17/02119), a "Precipita" Crowdfunding grant (FECYT). D.E. is funded by a Miguel Servet Fellowship (ISC III, CP12/03114, Spain); M.Z. is supported by a scholarship from Universidad Pública de Navarra; H.A. is supported by a scholarship from AECC; and M.G. is supported by a scholarship from the Government of Navarre.

### Author contributions

MZ designed and carried out experiments, collected data, and analyzed data. HA designed and carried out experiments, collected data, and analyzed data. GF-H recruited patients, collected data, and analyzed clinical data. MJG-G, MG, and AB carried out experiments, collected data, and analyzed data. MM, BH, LT, IM, MJL, AFL, and RV recruited patients, collected data, and analyzed clinical data. RV supervised the clinical staff, recruited patients, and analyzed clinical data. GK conceived the project, supervised non-clinical researchers, analyzed data, and wrote the paper. DE conceived the project, supervised non-clinical researchers, analyzed data, and wrote the paper. All authors participated in the writing of the manuscript.

### Conflict of interest

The authors declare that they have no conflict of interest.

### For more information

(i) https://www.cancerresearchuk.org/about-cancer/cancer-in-general/treatment/immunotherapy
(ii) https://www.aecc.es/es/red-social/testimonios/tratamientos-inmunoterapia
(iii) https://www.cancer.org/treatment/treatments-and-side-effects/treatment-types/immunotherapy.html

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
