## [Review Process File · EMBO Molecular Medicine]

Functional systemic CD4 immunity is required for clinical responses to PD-L1/PD-1 blockade therapy

Miren Zuazo, Hugo Arasanz, Gonzalo Fernández-Hinojal, Maria Jesus García-Granda, María Gato, Ana Bocanegra, Maite Martínez, Berta Hernández, Lucía Teijeira, Idoia Morilla, Maria Jose Lecumberri, Angela Fernández de Lascoiti, Ruth Vera, Grazyna Kochan, and David Escors

Review timeline:

Submission date:	9 February 2019
Editorial Decision:	25 February 2019
Revision received:	27 March 2019
Editorial Decision:	8 May 2019
Revision received:	10 May 2019
Accepted:	14 May 2019

Editor: Lise Roth

Transaction Report:

1st Editorial Decision

25 February 2019

Thank you for the submission of your manuscript to EMBO Molecular Medicine. We have now heard back from the two referees whom we asked to evaluate your manuscript.

As you will see from the reports below, while the referees mention the interest of the study, they also raise substantial concerns on your work, which should be convincingly addressed in a major revision of the present study. In particular, it will be crucial to improve the clarity of the manuscript, and further strengthen the mechanistic link between differentiated CD4 T cells and immune checkpoint responsiveness.

Addressing the reviewers' concerns in full (above points as well as other reviewers' comments) will be necessary for further considering the manuscript in our journal. Still, revising the manuscript according to the referees' recommendations appears to require a lot of additional work and experimentation. I am unsure whether you will be able or willing to address those and return a revised manuscript within the 3 months deadline. On the other hand, given the potential interest of the findings, I would be willing to consider a revised manuscript with the understanding that acceptance of the manuscript would entail a second round of review as some claims were not fully evaluated from the current data due to limitations highlighted by the referees (please see report from referee #1). As EMBO Molecular Medicine encourages a single round of revision only, and to save you from any frustrations in the end, I would strongly advise against returning an incomplete revision and would also understand your decision if you choose to rather seek rapid publication elsewhere at this stage.

EMBO Molecular Medicine has a "scooping protection" policy, whereby similar findings that are published by others during review or revision are not a criterion for rejection. Should you decide to submit a revised version, I do ask that you get in touch after three months if you have not completed it, to update us on the status. Please also contact us as soon as possible if similar work is published elsewhere. If other work is published, we may not be able to extend the revision period beyond three months.

I look forward to receiving your revised manuscript.

***** Reviewer's comments *****

Referee #1 (Remarks for Author):

the main concern I have is the lack of clarity throughout the manuscript. The topic is very interesting but not presented in a clear enough manner. The paper needs to be revised and rewritten to increase clarity. One example of many: It is not clear when the authors mention differences between G1 and G2 if they mean T cells or patients cohorts

Referee #2 (Remarks for Author):

In their manuscript entitled "Functional systemic CD4 immunity is required for clinical responses to PD-L1/PD-1 blockade therapy", Zuazo et al. examined the differential characteristics between responders and non-responders to PD-L1/PD-1 checkpoint blockade in lung cancer patients which had progressed on platinum-based chemotherapy. These clinical explorations lead the authors to identify a differential accumulation of highly-differentiated circulating CD4 (CD4HD) cells in NSCLC patients, with a subset of patients with greater accumulation of CD4HD responding well to immune checkpoint blockade. The authors subsequently divided patients into two cohorts based on higher abundance of CD4HD (G1) or lower abundance of CD4HD (G2) and performed analyses of T cell functionality. They found higher co-expression of PD-1 and Lag3 on G2 cells than G1 upon stimulation, with correspondingly lower proliferation and reduced responsiveness to treatment with anti-PD-1 as a single agent. They further demonstrated that these cells were not senescent cells, and found no difference in cytokine production between G1 and G2 in either highly differentiated or non-highly differentiated CD4. They further investigated these phenotypes in CD8 T cells and did not find similar differences in differentiation status or survival, but did observe a robust proliferative enhancement in CD8 T cells from G1 patients after therapy, specifically in the less highly differentiated (CD28+) subpopulation, and a similar higher co-expression of PD-1 and Lag3 on G2 cells than G1 upon stimulation. Encouragingly, while single agent anti-PD-1 could not enhance proliferation in cells from G2 patients, co-inhibition of PD-1 and Lag3 enhanced proliferation of both CD4+ and CD8+ cells from G2 patients. Finally, the authors show that combining G1 status with PD-L1 expression yielded the most robust metric yet shown to predict patient progression-free survival.

This manuscript details a functional characterization of circulating CD4 and CD8 T cells grouped based on differentiation status determined by CD27 and CD28 expression, and details an interesting link between proportion of highly-differentiated CD4 T cells and response to checkpoint blockade. However, concerns about the manuscript arise from a lack of a substantiated causal mechanistic link being demonstrated between their observation and immune checkpoint responsiveness. The fundamental differences demonstrated in the manuscript can arguably be traced back to the difference in inhibitory checkpoint expression between G1 and G2 patients, but without further depth of exploration. Major and minor concerns are as follows:

Major concerns:

- Concerned with the varying demarcation between G1 and G2 in the different datasets, making its actual utility tough to discern: in the validation dataset, the distribution of G1 and G2 was indistinguishable from the healthy donors in Figure 1A, with objective responders below the 40% mark in the validation dataset.
- Concerned with the analysis done in Supplemental Figure 2, where the conclusion was drawn that CD4 T cell profiling does not have significant prognostic value. This dataset seems to be drawn from the same dataset as Figure 1D, with the change that the responding patients have been removed from the analysis. It is unsurprising that following removal of all patients which responded to therapy and leaving only a comparison of the non-responding patients that a non-statistically

significant difference in survival remained. The analysis and conclusion seem contrived and misleading given those manipulations of the dataset and the direct comparison between that analysis and the analysis presented in Figure 1D.

- Main phenotype is that if fewer CD4HD, a larger proportion of circulating CD4 T cells is terminally exhausted (Figure 2) and can't respond to single-agent blockade: little cytokine production in either set. How does this observation of enhanced numbers of terminally-exhausted T cells relate to differentiation status? Inadequate exploration of this key aspect.

- Figure 3A is misleading- the data shown are not representative of the dataset, where the mean %TAA-specific CD4 T cells in both G1 and G2 quantified in Figure 3B are approximately 8% and 5%, while the samples shown in Figure 3A are 21% and 22% respectively. This difference is the difference between a CD4 population with a predominantly exhausted phenotype and a CD4 population with a relatively robust response.

- Wording in describing the mechanisms of dysfunction and ramifications of checkpoint inhibition are out of sync with current understanding of exhaustion biology. The authors repeatedly claim that PD-1 and Lag3 are driving the dysfunction phenotype- this is a controversial and at least partly inaccurate characterization of the biology (clarified well in Hashimoto et al, Annu Rev Med, 2018, and with in-depth analysis of CD4s in Crawford et al, Immunity, 2014). A discussion of and experimental probing of terminally exhausted vs. potentially reversible exhaustion in these various CD4 subsets is required.

- Incomplete discussion of the data in the context of previously published literature- Spitzer et al (Cell, 2017) demonstrated clearly the requirement of systemic immunity for effective cancer immunotherapy, while Markowitz et al (JCI Insight, 2018) demonstrated the importance of both CD4 and CD8 T cells in NSCLC response to checkpoint blockade.

Minor concerns:

- Axis title wrong in Figure 4A- says CD4 T cells when discussing CD8 T cells.

- Methods section seems incomplete from antibodies being listed in the Flow Cytometry section but without further details.

1st Revision - authors' response

27 March 2019

Reviewer 1.

“The main concern I have is the lack of clarity throughout the manuscript. The topic is very interesting but not presented in a clear enough manner. The paper needs to be revised and rewritten to increase clarity. One example of many: It is not clear when the authors mention differences between G1 and G2 if they mean T cells or patients cohorts “

The main concern of Reviewer 1 was the clarity of the paper. It is true that the manuscript needs the clarifications as pointed by Reviewer 1. Therefore, to improve the clarity of the paper as suggested by Reviewer 1 we have done a major revision of our paper, not only addressing his/her main point, but also the issues raised by Reviewer 2. These are too numerous to list, and are highlighted in yellow in the main text which covers most of the current text. We also confirm that we have clarified in the relevant places when we talked about T cells or G1/G2 cohorts. As a result, we have a much clearer and simpler manuscript.

Reviewer 2.

Reviewer 2 has thoroughly revised the paper, and reading his summary of our manuscript and many of the points on cytokine production and T cell exhaustion we have realized that he/she misunderstood some of the figures and data. However, this has made us to perform numerous new experiments and include additional data to show that CD4 T cell systemic dysfunctionality is not caused by T cell exhaustion or bona fide T cell anergy, but specifically on proliferative dysfunctionality, which was reversible by PD1/LAG3 co-blockade. The new data is now presented as Figure 4 and some new data in Expanded View Figures as mentioned in each addressed point, below.

Therefore, we have addressed all the concerns raised by the reviewer, point by point, as follows.

Major concerns:

** Concerned with the varying demarcation between G1 and G2 in the different datasets, making*

its actual utility tough to discern: in the validation dataset, the distribution of G1 and G2 was indistinguishable from the healthy donors in Figure 1A, with objective responders below the 40% mark in the validation dataset.

We understand the concern of Reviewer 2, as we did not sufficiently explain the differences between the two datasets and why that difference is certainly expected. Data from the discovery cohort was obtained by direct T cell staining from freshly isolated PBMCs within the same day of sample retrieval. Data from the validation cohort was generated by an independent researcher using a drastically different protocol based on (1) overnight depletion of myeloid cells by adherence to plastic. (2) Recovery of non-adherent cells after overnight depletion of myeloid cells. (3) T cell staining from remaining non-adherent cells after the overnight incubation, followed by flow cytometry.

Hence, the cut-off values used to differentiate G1 and G2 patients vary between the two cohorts. Indeed, the difference is close to a “mathematical” 20%. As we stated in the discussion, our intention was not to provide a fully validated biomarker of responses for direct clinical application. The correlation between the two procedures was nearly perfect, considering a cut-off value for the validation cohort of 20% which was corroborated by ROC analysis as shown in the Expanded View Figure. To apply this biomarker to the clinic, a standardised protocol has to be established for sample retrieval, processing and analysis. All diagnostic/analytical clinical tools have to be validated always with a standard protocol that has to be followed “by the book” to ensure reproducibility for clinical use.

Considering the major differences in protocols, it is not appropriate to compare the percentages from Expanded View Figure 1 with the data in healthy donors from Figure 1. The two protocols are too different. Hence, we have added new data from healthy donors using the validation protocol and added these data in expanded view figure 1. This comment has made us realize that this point has to be made clearer in the text, and we have amended and hopefully satisfactorily addressed Reviewer 2’s concerns as follows:

In page 3, the differences between the two protocols have been explained in detail as follows “While in the discovery cohort T cells were directly analyzed from peripheral blood samples within the same day, validation samples were processed very differently. Briefly, an overnight depletion step of myeloid cells by adherence to plastic was included before T cell analyses from non-adherent cells. Hence, relative percentages of CD4 T_{HD} cells varied between the discovery and validation cohorts. Even so, there was a significant agreement between the two datasets on patient classification as demonstrated by Cohen’s kappa coefficient ($\kappa=0.932$). The highly significant association between G1 patients and objective responses in the validation set was confirmed ($P=0.0006$), albeit with a cut-off value of 20% in the validation dataset which was corroborated by ROC analysis (Fig EV1).

In agreement with these results, the G1 patient cohort had a significantly longer progression-free survival (PFS) compared to the G2 cohort.”

- In the Expanded View Figure 1 itself, we have incorporated quantification of CD4 THD cells from healthy donors using the protocol of the validation dataset so that now these percentages can be compared without bias, and not with data from Figure 1A. As expected, now the values from healthy donors have decreased correspondingly. We have amended the figure legend as follows:

“**Expanded figure 1. Validation dataset.** (A) Distribution of circulating CD4 T_{HD} cells within CD4⁺ CD14^{negative} cells in healthy donors (N=14) and in NSCLC patients constituting the validation set (N=32). G1 and G2 groups are indicated and separated by the mean (horizontal line). The means \pm standard deviations of CD4 T_{HD} cells in G1 and G2 groups are shown on the right, as well as the association between G1 profiles and objective responses by the Fisher’s test. Differences between healthy donors and NSCLC patients were tested with the U of Mann-Whitney test. (B) ROC analysis of CD4 T_{HD} quantification in the validation dataset and objective responses. The cut-off value for identification of responses is shown in the graph. *, **, indicate significant ($P<0.05$) and highly significant ($P<0.001$) differences.”

- In Discussion, we have commented again on these differences and the importance of using a standard protocol for application to clinical practice, in page 7 as follows: “It is worth noting that the cut-off value was reduced to 20% in a validation cohort that was independently processed and analysed by a very different procedure. Importantly, patient classification in G1 or G2 cohorts and their association with clinical responses agreed independently of the protocol utilized. We are well aware that quantification of CD4 T_{HD} cells could be used as a baseline factor for clinical stratification. Proper validation of CD4 T cell profiling will require protocol standardisation for sample manipulation and analyses.
- The differences between the two protocols have been highlighted in Materials and Methods as follows:

“Validation dataset

Data from a set of 32 patients was validated in parallel by independent handling, processing, staining, flow cytometry data collection and analysis. The validation dataset was generated by a technician working in unrelated research themes (A.B.). A very different protocol was used to quantify CD4 T_{HD} cells in the validation set compared to the discovery cohort. For the validation dataset, isolated PBMCs were resuspended in TexMacs serum-free medium (Miltenyi) and plated on 6-well cell culture plates. Myeloid cells were allowed to adhere overnight, and non-adherent cells were collected, centrifuged, resuspended and T cells stained with the appropriate antibodies for flow cytometry analyses. ROC analysis was used to establish the cut-off value for the relative percentage of CD4 T_{HD} cells to discriminate G1 vs G2 patients in the validation cohort. *Post hoc* Cohen’s kappa coefficient test was used to test the agreement between the discovery cohort versus the validation cohort on classification of G1/G2 patients. “

** Concerned with the analysis done in Supplemental Figure 2, where the conclusion was drawn that CD4 T cell profiling does not have significant prognostic value. This dataset seems to be drawn from the same dataset as Figure 1D, with the change that the responding patients have been removed from the analysis. It is unsurprising that following removal of all patients which responded to therapy and leaving only a comparison of the non-responding patients that a non-statistically significant difference in survival remained. The analysis and conclusion seem contrived and misleading given those manipulations of the dataset and the direct comparison between that analysis and the analysis presented in Figure 1D.*

We apologise to the Reviewer for not having explained in more depth the reasoning for the analysis behind the now Expanded View Figure 2. We have to stress that no dataset has been manipulated in anyway. All the data that we obtain is the data that we show and that describe in the Figure legends. All the data from the manuscript comes from the same patients, and therefore from the same dataset as in Figure 1D, as in Figure 1A, as in Figure 5, etc. We never claimed this to be different either in the figure legend, in the text or in materials and methods. Our intention was not to mislead anyone, but possibly wording for data presentation may have unintentionally given that impression from our part. We apologise for that.

The analysis shown in the now Expanded View Figure 2 is the result of intense discussion with our oncologists. According to them, if G1 profiles have good prognostic properties, G1 non-objective responders would still show some benefit compared to G2 patients, possibly in the form of stable disease or longer survival. For them it was important to remove objective responders in order not to confound the prognostic value to be analysed. I agree that for many scientists this reasoning might not be that logical. As we do not want to give any impression of misleading, we have now used a more conventional procedure to assess prognosis in immunotherapy, previously used for anti-PD-1 immunotherapies in top clinical papers (For example, in Le et al. NEJM. 2015. *PD-1 Blockade in Tumors with Mismatch-Repair Deficiency*). This is based on the analysis of the relative time elapsed from diagnosis to recruitment to immunotherapy. Hence, if patients with a G1 profile have better prognosis than G2, we would expect longer periods of time from the moment of diagnosis to the start of immunotherapies. As this is estimated before the start of immunotherapies, it is not confounded by the therapy itself. As our patients have received similar conventional chemotherapies, the treatments before immunotherapy would not affect this either. Therefore, we have decided to replace this figure by a more conventional one based on time from diagnosis to enrolment in immunotherapies. The conclusion is the same, but now data from all patients is included. This

analysis discards any significant prognostic value. Not that it would change the core of the paper if it would have.

Finally, the conclusion of prognosis was not exclusively drawn from that figure, but also from correlation studies of CD4 T cell profiling and other well-established prognostic variables, as shown in the manuscript.

Overall, CD4 T cell profiling could still have prognostic value (it would be equally important to our paper), and not change anything in the core of our manuscript. However, this is not the case and it does not have prognostic value. That is the reason why this data is going as Expanded View data and not in the main manuscript.

Therefore, to clarify this issue, we have replaced the now Expanded View Figure 2 and changed the manuscript as follows:

- In results, page 4: “To assess whether CD4 T cell profiling had prognostic value, the time elapsed from diagnosis to the start of immunotherapies was compared between G1 and G2 patient cohorts, as described (Le et al, 2015). No significant differences were observed, indicating that G1/G2 classification did not have prognostic value (Fig EV2). This was supported by no association between G1/G2 patient cohorts and baseline ECOG score (P=0.6), with liver metastases (P=0.88), with tumor load (P=0.19) or with the Gustave-Roussy immune score (GRIm) (P=0.14, Table EV2) (Bigot et al, 2017).
- Figure legend from supplementary figure 2: “**Expanded Figure 2. CD4 T cell profiling does not have significant prognostic value. Kaplan-Meier plot of relative time elapsed from diagnosis to the start of immunotherapy for G1 (blue) and G2 patient cohorts (red), as indicated. No significant differences were found.**”
- **Main phenotype is that if fewer CD4HD, a larger proportion of circulating CD4 T cells is terminally exhausted (Figure 2) and can't respond to single-agent blockade: little cytokine production in either set. How does this observation of enhanced numbers of terminally-exhausted T cells relate to differentiation status? Inadequate exploration of this key aspect.**

This is probably the key aspect raised by the Reviewer. Here we need to point out to the Reviewer that nowhere in Figure 2 we have shown any cytokine production by any T cell, or T cell subset or assessed their exhausted, or anergic state. We believe that the Reviewer mistook some of the data from the panels to be cytokine production, and assumed that we were showing that these T cells were terminally-exhausted cells. All the data shown in these panels relate to proliferation without any more insight.

Hence, the Reviewer mistakenly thought that we were suggesting that CD4 T cells from G2 patients were indeed terminally exhausted T cells or their relationship with differentiation status according to CD28 expression. Nowhere in the manuscript had we explored the specific nature of the T cell dysfunctionality apart from proliferative capacities. Not surprisingly, the Reviewer is at a loss on how “terminally-exhausted T cells” relate to T cells from our patient cohorts or to their differentiation.

Indeed, we agree that this is a key aspect and we did not explore it sufficiently. The data from this aspect also extends to other comments from Reviewer 2. Therefore, we have employed a considerable effort to perform experiments to solve this key issue, and generated new data to clarify the nature of T cell dysfunctionality in our patient cohorts.

As the reviewer is aware, there is much confusion even in the specialized literature regarding the characteristics and differences between exhausted T cells, anergic T cells and senescent T cells. Accordingly, the reviewer provided some literature in which T cell exhaustion is reviewed and characterized (Hashimoto et al; Crawford et al). T cell exhaustion is a common phenomenon in some chronic viral infections, by which through a prolonged and repeated T cell stimulation with heterologous antigens (usually high-, medium-affinity TCR activation), T cells go through an “overstimulation” state that constitutively triggers negative feedback mechanisms. As a consequence, these T cells exhibit high constitutive surface expression of immune checkpoints such as PD-1, LAG3 or TIM-3. But the defining characteristic is the progressive loss of cytokine production, and the loss of T cell multifunctionality (the capacity for multi-cytokine production, and

hence for plasticity of differentiation towards a range of T helper phenotypes). Proliferation in these T cells is compromised.

The most common dysfunctionality in cancer is T cell anergy, which is triggered by suboptimal antigen presentation to T cells (lack of signals 2 and 3), or by the presence of high levels of immunosuppressive cytokines during antigen presentation (IL10, TGF-beta, etc). This indeed constitutes a central tolerogenic mechanism to prevent T cell attack towards autoantigens and hence, most tumor-associated antigens. Although the phenotype of anergic T cells is varied, the key characteristic is that these cells undergo one round of expansion following the first recognition of antigen, but fail to proliferate after subsequent stimulation. Anergic T cells do generally (but not always) express lower levels of cytokines (particularly IL2). In contrast to exhausted T cells, anergic T cells do not *constitutively express* high surface levels of PD1, LAG3 or TIM3. Moreover, T cell anergy is easily reversible by cytokine re-priming with IL2, IL15 or IL12.

We agree with the Reviewer that characterizing the nature of T cell dysfunctionality in non-responder patients is paramount, and linked to other requests by Reviewer 2. We have generated a considerable amount of new data that is shown in the newly incorporated Figure 4. Briefly, CD4 CD28positive T cells from G2 patients are not exhausted and they even have significant capacities for multicytokine production. Neither are the CD28^{negative} counterparts. We found that CD28⁺ CD4 T cell dysfunctionality in G2 patients is restricted to proliferative unresponsiveness coupled to strong PD-1/LAG-3 co-upregulation following stimulation. Our results show that these T cells have a certain degree of anergy that does not compromise their capacities to produce cytokines. Apart from this, we have confirmed the Th17-skewed CD4 responses typical of lung cancer patients. We have added our conclusions in the manuscript in results and discussion as follows:

- Page 4, results. The referee points out that the main phenotype is that the larger proportion of highly-differentiated subsets, the lower proportion of “exhausted” (dysfunctional in proliferation, as our new data shows now) CD4 T cells there are. In general terms, this is the main finding. To make sure that it is clear throughout the text, we have first strengthened the fact that THD cells are not exhausted, anergic or senescent cells due to their high proliferative capacities, as follows: “The strong proliferative capacities of CD4 T_{HD} cells indicated that these were not exhausted, anergic, or senescent subsets, but probably highly differentiated memory subsets. To test this, their baseline phenotype according to CD62L/CD45RA surface expression was assessed in a sample of patients (Fig EV4A)”.
- Page 5, results: “Our results strongly suggested that circulating CD4 T_{HD} cells in our cohort of NSCLC patients mostly corresponded to non-senescent, non-exhausted memory subsets.”

Then, we have addressed the nature of CD4 T cell dysfunctionality in CD28 positive subsets in G2 patients compared to G1 patients and healthy donors as follows:

- Page 5-6, results: “To further study the dysfunctional status of systemic CD4 T cells in G2 patients, we evaluated PD-1 and LAG-3 surface expression directly after blood sampling, as constitutive high level expression of these markers is a frequent characteristic of T cell exhaustion. However, no differences were found between age-matched healthy donors and G1/G2 patient cohorts in either CD28⁺ or CD28^{negative} subsets (**not shown**). Nevertheless, the defining hallmark of T cell exhaustion is the loss of cytokine production following stimulation, particularly multicytokine expression (Crawford et al, 2014). Interestingly, CD4 T cells from both G1 and G2 patient cohorts were as proficient in IFN- γ , IL4, IL10 and IL2 expression as T cells from healthy donors independently of their CD28 expression (Fig 4A). Indeed, CD4 cells (total, CD28⁺ and CD28^{negative} subsets) in both G1 and G2 patient cohorts were significantly skewed towards IL17 responses compared to age-matched healthy donors (Fig 4A). Importantly, only a minority of CD4 T cells from either G1 or G2 patient groups were single-cytokine producers (Fig 4B) while most of the CD28⁺ CD4 T cells were very proficient in multiple cytokine production with a preference for IL17-expressing subsets (Fig 4C-D). These results indicated that CD4 T cells from G2 patients were not exhausted according to our current understanding (Hashimoto et al, 2018). Indeed, they responded to stimulation by producing cytokines although with strong

co-upregulation of PD-1/LAG-3 associated with markedly diminished proliferative capacities.”

- Page 7. Discussion. “Systemic CD28⁺ CD4 T cells in G2 patients were not truly exhausted or *bona fide* anergic T cells. No constitutive high-level expression of PD-1 and LAG-3 was observed unless stimulated. They were proficient in multi-cytokine expression following stimulation. Indeed, CD4 T cells from both G1 and G2 patient cohorts were skewed towards Th17-expressing phenotypes compared to healthy donors. All these characteristics were indicators of systemic CD4 T cell proliferative dysfunctionality in G2 patients.”

*** Figure 3A is misleading- the data shown are not representative of the dataset, where the mean %TAA-specific CD4 T cells in both G1 and G2 quantified in Figure 3B are approximately 8% and 5%, while the samples shown in Figure 3A are 21% and 22% respectively. This difference is the difference between a CD4 population with a predominantly exhausted phenotype and a CD4 population with a relatively robust response.**

We sincerely apologise to the Reviewer. He is right, the data shown in Figure 3A is not representative of the dataset as described in the figure legend. We did not intend to claim it was. It has been a case of “bad wording” in the figure legend. Our intention was not to mislead. In fact, we do show all the data in Figure 3B without any major issues, and we concluded in our manuscript exactly the same conclusion drawn by the Reviewer (“These results suggested that poor responses in G2 patients were not caused by lack of tumor-specific CD4 T cells but rather by having dysfunctional T cells.”). We meant to say that the FACS plots were intended to represent the technical quality of the data from our antigen presentation assay for detecting lung cancer-specific T cells. Nothing more than that. There is a tendency nowadays in the papers to provide all the data in graphs without showing primary data directly from FACS plots, for example. In this situation it is difficult to judge if compensation has been performed correctly, the number of cells that have been truly analysed for conclusions (“few dots” vs “large cloud of events”) and how negative controls look like. I am sure Reviewer 2 understands this. Hence, in most of our papers we like to present primary data in the form of FACS plots rather than provide only “processed” data in graphs. Our manuscript is full of examples.

Nevertheless, to avoid any misunderstanding or unintentional misleading, we decided to simply remove panel A in Figure 3. We have therefore changed Figure 3 panels B and C to panels A and B, and changed their referencing in the manuscript and in the figure legend as follows:

- Results, page 5. “Then we thought that G2 patients could be refractory to anti-PD-1 immunotherapy by not having systemic cancer-specific CD4 T cells. To this end, we quantified CD4 T cells reactive to lung adenocarcinoma antigens using IFN- γ -activated autologous monocyte-derived DCs as antigen presenting cells, as described (Escors et al, 2008). DCs were loaded with A549 cell lysate, as these cells contain numerous common lung adenocarcinoma antigens (Madoz-Gurpide et al, 2008). We used this approach as we lacked sufficient biopsy material to get tumor antigens or tumor-infiltrating T cells. CD4 T cells reactive to A549 cell antigens were identified by IFN- γ upregulation. Interestingly, lung cancer-specific CD4 T cells were present at varying proportions before the start of immunotherapy in both G1 and G2 patients (Fig 3A). Indeed, percentages of lung cancer-specific CD4 T cells did not differ significantly between G1 (responders and non-responders) and G2 patients. These T cells consisted of both T_{HD} and non-T_{HD} subsets, without significant differences in relative percentages between G1 and G2 cohorts (Fig 3B). These results suggested that poor responses in G2 patients were not caused by lack of tumor-specific CD4 T cells but rather by having dysfunctional T cells.”
- The legend to Figure 3 was correspondingly changed as follows: “**Figure 3. Lung cancer antigen-specific CD4 T cells in NSCLC patients.** (A) Scatter plot graph with the percentage of lung cancer-specific systemic CD4 T cells quantified by an autologous DC-based antigen presentation assay (See Materials and Methods), in a sample of G1 and G2 patients as indicated. Objective responses (OR) are shown in green. In red, patients with no OR. (B) The scatter plot graph on the left represents the percentage of CD4 T_{HD} cells within lung-cancer specific CD4 T cells in a sample of patients from the indicated G1/G2

groups. On the right, same as left but representing the percentage of CD28⁺ CD4 T cells within lung-cancer specific CD4 T cells. Objective responders (OR) are shown in green. In red, patients with no OR. Relevant statistical comparisons are shown within the graphs with the test of Mann-Whitney. N, number of biological replicates (independent patients); Ns, no significant differences (P<0.05).”

** Wording in describing the mechanisms of dysfunction and ramifications of checkpoint inhibition are out of sync with current understanding of exhaustion biology. The authors repeatedly claim that PD-1 and Lag3 are driving the dysfunction phenotype- this is a controversial and at least partly inaccurate characterization of the biology (clarified well in Hashimoto et al, Annu Rev Med, 2018, and with in-depth analysis of CD4s in Crawford et al, Immunity, 2014). A discussion of and experimental probing of terminally exhausted vs. potentially reversible exhaustion in these various CD4 subsets is required.*

The Reviewer points out that wording on describing the mechanisms of dysfunction in our paper in light of current understanding of exhaustion biology needs to be changed. That we repeatedly claim that PD-1 and LAG3 are driving the dysfunction phenotype.

We thank to the Reviewer for this observation, and also his/her previous suggestions on T cell exhaustion. As shown above, we did extensive experimental work to address the previous issue and also this one. It turned out that CD4 T cells from G2 patients (both CD28⁺ and CD28^{negative} subsets) were not terminally exhausted, and not even exhausted at all. Therefore we cannot even apply the current knowledge of PD-1 and LAG-3 over T cell exhaustion. We agree that our claim that PD-1/LAG-3 are driving the dysfunction phenotype is an overstatement. Indeed, CD4 T cells from G2 patients up-regulate PD-1/LAG-3 AFTER stimulation, but surely they contribute to inhibiting T cell proliferation in light of our co-blockade studies (original Figure 4F and 4G, now Figure 6). Therefore, co-upregulation of PD-1/LAG-3 is a consequence rather than the cause. We have amended and clarified this fact throughout the manuscript as shown below. The Reviewer has provided two excellent papers (a review and a scientific paper) in which they clarify the nature of exhausted T cells. The tested characteristics to check for T cell exhaustion in our paper have been referenced to these two papers in the results section. They provide very useful information for the reader. This also has been amended in the text as will be shown below.

Because the Reviewer originally thought that we were having exhausted T cells, she/he encouraged us to discuss whether we were observing terminally exhausted or reversible exhaustion in our T cells, and to experimentally test this over our CD4 T cells. Again, we performed experiments that showed that CD4 T cells from G2 patients were dysfunctional only in proliferation but not in cytokine production, independently of their CD28 expression. We cannot therefore test whether we have terminal exhaustion or reversible exhaustion as originally suggested as these are not exhausted cells, but we did show that proliferative dysfunctionality is reversible over CD4 T cells from G2 patients independently of their CD28 status by co-blockade of PD-1/LAG-3 but not by single blockade of either PD-1 or LAG-3. As these CD4 T cells are very proficient in multicytokine expression, we cannot test for the recovery of cytokine production. We have additional data that we are saving for another paper in which we show that CD4 T cells from G2 patients recover proliferative capacities also by cytokine priming, which reinforces the notion of these cells having a degree of T cell energy that can be reversed by proper priming. However, we would prefer not to include this data at this stage here unless the Referee disagrees. It would not change the core of the paper.

To address the overstatement on PD-1/LAG-3 co-expression driving dysfunctionality, we have added the following:

- We have changed in the abstract the phrase “T cell proliferative dysfunctionality was caused by PD-1/LAG-3 co-expression, and could be reverted by co-blockade” by “T cell proliferative dysfunctionality could be reverted by PD-1/LAG-3 co-blockade.”
- In the abstract we have commented on their capacities to produce cytokines as follows: “Although proficient in cytokine production, CD4 T cells in these patients proliferated very poorly, strongly co-upregulated PD-1/LAG-3, and were largely refractory to PD-1 monoblockade.”

- In Introduction, we have changed the phrase “Our results indicate that baseline functional systemic CD4 immunity is required for objective clinical responses to PD-L1/PD-1 blockade therapies, with PD-1/LAG-3 co-expression causing CD4 and CD8 dysfunctionality in non-responder patients.” by “Our results indicate that baseline functional systemic CD4 immunity is required for objective clinical responses to PD-L1/PD-1 blockade therapies.”
- Results, page 5, “CD4 T cells of G2 patients strongly co-upregulated PD-1/LAG-3 after stimulation. We wondered if lack of clinical responses in G2 patients could be explained by resistance to single blockade of PD-1.”
- Results, page 6, the following sentence “As we found that CD4 dysfunctionality correlated with high PD-1/LAG-3 co-expression in G2 patients, we tested if this was also the case for CD8 T cells” was changed to “As we found that CD4 proliferative dysfunctionality in G2 patients correlated with high PD-1/LAG-3 co-upregulation after activation, we tested if this was also the case for CD8 T cells.”
- Results, page 6. We changed the text to the following : “**Proliferative dysfunctionality of CD4 and CD8 T cells from G2 patients is reversible after PD-1/LAG-3 dual blockade.** As we found that CD4 proliferative dysfunctionality in G2 patients correlated with high PD-1/LAG-3 co-upregulation after activation, we tested if this was also the case for CD8 T cells. PD-1/LAG-3 co-expression was tested *ex vivo* after stimulation with A549-SC3 cells, and G2 patients presented a significantly higher proportion of PD-1/LAG-3 co-expressing CD8 T cells compared to G1 counterparts (**Fig 6A**). Overall, our data indicated that PD-1/LAG-3 co-upregulation was contributing to proliferative dysfunctionality. To test if this was the case, baseline samples of CD4 and CD8 T cells from G2 patients were co-incubated *ex vivo* with A549-SC3 cells in the presence of an isotype antibody control, anti-PD-1, anti-LAG-3 or anti-PD1/anti-LAG-3 antibodies. We confirmed that each antibody was specifically blocking PD-1, LAG-3 or both in our assays by epitope masking using flow cytometry (**not shown**). Only co-blockade of PD-1 and LAG-3 in both CD4 (**Fig 6B**) and CD8 T cells (**Fig 6C**) from G2 patients significantly increased proliferation independently of CD28 expression. These results confirmed that PD-1/LAG-3 co-upregulation contributed to keeping systemic CD4 and CD8 T cells from G2 patients in a proliferative dysfunctional state following stimulation, and that this T cell dysfunctionality can be reverted by co-blockade of both immune checkpoints.”
- Figure (4) now (5) legend. We have changed it as follows; “**CD8 dysfunctionality recovers in G1 patients undergoing immunotherapy....**”

Referencing in the text to characteristics of T cell exhaustion from Hashimoto et al. Annu Rev Med. 2018 and Crawford et al. Immunity 2014.

- Results, page 5: “Nevertheless, the defining hallmark of T cell exhaustion is the loss of cytokine production following stimulation, particularly multicytokine expression (Crawford et al, 2014). Interestingly, CD4 T cells from both G1 and G2 patient cohorts were as proficient in IFN- γ , IL4, IL10 and IL2 expression as T cells from healthy donors independently of their CD28 expression (**Fig 4A**). Indeed, CD4 cells (total, CD28⁺ and CD28^{negative} subsets) in both G1 and G2 patient cohorts were significantly skewed towards IL17 responses compared to age-matched healthy donors (**Fig 4A**). Importantly, only a minority of CD4 T cells from either G1 or G2 patient groups were single-cytokine producers (**Fig 4B**) while most of the CD28⁺ CD4 T cells were very proficient in multiple cytokine production with a preference for IL17-expressing subsets (**Fig 4C-D**). These results indicated that CD4 T cells from G2 patients were not exhausted according to current knowledge (Hashimoto et al, 2018).”

To address the discussion and testing of terminally exhausted versus reversible exhaustion in our T cells, we experimentally assessed whether they were truly exhausted T cells. They were not, and we have added a considerable amount of new data included in a new figure (Figure 4). The manuscript has been changed by adding the following:

- Results, page 5: “To further study the dysfunctional status of systemic CD4 T cells in G2 patients, we evaluated PD-1 and LAG-3 surface expression directly after blood sampling, as constitutive high level expression of these markers is a frequent characteristic of T cell exhaustion. However, no differences were found between age-matched healthy donors and G1/G2 patient cohorts in either CD28⁺ or CD28^{negative} subsets (**not shown**). Nevertheless, the defining hallmark of T cell exhaustion is the loss of cytokine production following stimulation, particularly multicytokine expression (Crawford et al, 2014). Interestingly, CD4 T cells from both G1 and G2 patient cohorts were as proficient in IFN- γ , IL4, IL10 and IL2 expression as T cells from healthy donors independently of their CD28 expression (**Fig 4A**). Indeed, CD4 cells (total, CD28⁺ and CD28^{negative} subsets) in both G1 and G2 patient cohorts were significantly skewed towards IL17 responses compared to age-matched healthy donors (**Fig 4A**). Importantly, only a minority of CD4 T cells from either G1 or G2 patient groups were single-cytokine producers (**Fig 4B**) while most of the CD28⁺ CD4 T cells were very proficient in multiple cytokine production with a preference for IL17-expressing subsets (**Fig 4C-D**). These results indicated that CD4 T cells from G2 patients were not exhausted according to current knowledge (Hashimoto et al, 2018). Indeed, they responded to stimulation by producing cytokines although with strong co-upregulation of PD-1/LAG-3 associated with markedly diminished proliferative capacities.”
- Discussion, page 7: “Systemic CD28⁺ CD4 T cells in G2 patients were not truly exhausted or *bona fide* anergic T cells. No constitutive high-level expression of PD-1 and LAG-3 was observed unless stimulated. They were proficient in multi-cytokine expression following stimulation. Indeed, CD4 T cells from both G1 and G2 patient cohorts were skewed towards Th17-expressing phenotypes compared to healthy donors. All these characteristics were indicators of systemic CD4 T proliferative dysfunctionality in G2 patients”.

Regarding the reversibility of the “exhaustion” phenotype, as we did not have exhausted T cells, we had already shown in the original Figure 5 (now Figure 6) that this proliferative dysfunctionality could be reverted by PD-1/LAG-3 co-blockade independently of their CD28 status. Moreover, we have additional data for another paper in preparation in which we show the reversibility by cytokine priming. Nevertheless, we do not think this data on its own will add significant novel information in the context of this paper, and we would rather keep it for the follow-up. Unless the Reviewer considers otherwise. The data and the explicit reversion of the proliferative dysfunctionality has been introduced as follows:

- In Results, page 6, on the reversibility of proliferative dysfunctionality:

“**Proliferative dysfunctionality of CD4 and CD8 T cells from G2 patients is reversible after PD-1/LAG-3 dual blockade.** As we found that CD4 proliferative dysfunctionality in G2 patients correlated with high PD-1/LAG-3 co-upregulation after activation, we tested if this was also the case for CD8 T cells. PD-1/LAG-3 co-expression was tested *ex vivo* after stimulation with A549-SC3 cells, and G2 patients presented a significantly higher proportion of PD-1/LAG-3 co-expressing CD8 T cells compared to G1 counterparts (**Fig 6A**). Overall, our data indicated that PD-1/LAG-3 co-upregulation was contributing to proliferative dysfunctionality. To test if this was the case, baseline samples of CD4 and CD8 T cells from G2 patients were co-incubated *ex vivo* with A549-SC3 cells in the presence of an isotype antibody control, anti-PD-1, anti-LAG-3 or anti-PD1/anti-LAG-3 antibodies. We confirmed that each antibody was specifically blocking PD-1, LAG-3 or both in our assays by epitope masking using flow cytometry (**not shown**). Only co-blockade of PD-1 and LAG-3 in both CD4 (**Fig 6B**) and CD8 T cells (**Fig 6C**) from G2 patients significantly increased proliferation independently of CD28 expression. These results confirmed that PD-1/LAG-3 co-upregulation contributed to keeping systemic CD4 and CD8 T cells from G2 patients in a proliferative dysfunctional state following stimulation, and that this T cell dysfunctionality can be reverted by co-blockade of both immune checkpoints.”
- In discussion, page 8 “Both CD4 and CD8 proliferative dysfunctionality in G2 patients was reversible *ex vivo* by PD-1/LAG-3 co-blockade.”

**** Incomplete discussion of the data in the context of previously published literature- Spitzer et al (Cell, 2017) demonstrated clearly the requirement of systemic immunity for effective cancer immunotherapy, while Markowitz et al (JCI Insight, 2018) demonstrated the importance of both CD4 and CD8 T cells in NSCLC response to checkpoint blockade.***

We sincerely thank the Reviewer for pointing these two papers to us. The paper in Cell by Spitzer et al, shows that functional immunity is required for one type of immunotherapy (which is not PDL1/PD1 blockade) in murine models. This is a high-quality paper, which it is based on the visualization of immune cell lineages across tissues using a new method of organizing high-throughput phenotypic data. In this paper they mainly administer tumor-binding immunoglobulins, not immune checkpoint inhibitors. Therefore, it cannot be claimed that the authors have truly demonstrated the requirement of systemic immunity for all immunotherapies. The conclusion is valid for that one type of immunotherapy in mice. The detailed mechanisms between immune checkpoint blockade and tumor-binding immunoglobulin administration on efficacy could be (or not) very different.

Interestingly the authors describe that the expansion of a specific systemic subset of murine CD4 T cells correlates with efficacy. As it turns out, *their data is in agreement with our data, but we believe there are important differences that bring value to our data.* First, as far as we are aware, our manuscript is the first to show in human patients undergoing clinical PDL1/PD1 blockade the requirement for functional systemic CD4 immunity. Second, care should be taken when translating results from murine models to human patients. Only one example, human CD4 T cell subsets according to CD27-CD28 expression profiles do not correspond to their phenotypic counterparts in murine T cells. This is only just one example of many. Third, *the prediction capabilities* of our data in human patients, at least in the context of NSCLC patients progressing from conventional therapies. We also are of the opinion that this brings additional value and interest to our data.

The recent paper by Markowitz et al in JCI insight is truly a top paper. They thoroughly analyse CD4 and CD8 T cells, but again, in murine models with the shortcomings (and advantages) that mouse immunology brings into the picture. Nevertheless, their data strongly agrees with our data in human patients. Therefore, both papers do reinforce our data, with our contribution of having done this study in human NSCLC patients undergoing PDL1/PD1 blockade.

We agree with the Reviewer that we must discuss our data in light of the results from these two papers, and we have done so as follows:

Discussion, page 8: “The requirement of functional systemic immunity has been previously demonstrated in murine models for the efficacy of other immunotherapy approaches (Spitzer et al, 2017), as well as the importance of CD4 T cells for anti-PD-1 immunotherapy (Markowitz et al, 2018). These studies are in agreement with our present data in human patients undergoing PD-L1/PD-1 blockade therapies. Indeed, the appearance of a specific murine subtype of CD4 T cell was the main correlator with efficacious responses by administration of anti-cancer cell immunoglobulins (Spitzer et al, 2017). These results together with our data strongly support the need for proficient CD4 responses to achieve efficacious responses.”

Minor concerns:

- **Axis title wrong in Figure 4A- says CD4 T cells when discussing CD8 T cells.**
We thank the reviewer for noticing, and it has been amended. Now it is Figure 5A.
- **Methods section seems incomplete from antibodies being listed in the Flow Cytometry section but without further details.**
We thank the reviewer for noticing, and we have included all information on the antibodies, their dilutions and also the new antibodies that we have used to carry out the experiments suggested by the Reviewer. All changes have been highlighted.

Thank you for the submission of your revised manuscript to EMBO Molecular Medicine. Please accept my apologies for the unusual delay in getting back to you, which is due to the fact that one referee was not responsive despite several chasers, and that I therefore sought external advice from a good expert in the field to reach a fair and balanced decision on your manuscript.

Both this adviser and referee #2 are now supportive of publication. I am therefore pleased to inform you that we will be able to accept your manuscript once the minor editorial amendments have been completed.

Referee and adviser reports:

No further experiment is required at that point; please address the concerns of referee #2 and of the adviser in writing.

The adviser stated:

"I think the manuscript is suitable for publication in EMM, I would have liked to see the population of G1 in figure 4 divided into responders and non-responders (or individual coloured dots as the other figures) just to see if there was a difference covered by the non responders especially in IFN- γ response." If feasible within a reasonable timeframe, please address this comment.

I look forward to reading a new revised version of your manuscript as soon as possible.

***** Reviewer's comments *****

Referee #2 (Remarks for Author):

The paper is much better than it was originally. Most of my concerns have been addressed, and I appreciate the new data and new transparency in discussing their results. The analysis concerning prognostic value has now been performed more robustly and with transparency in terms of correlations to different phenotypes (such as immune score and tumor load), and I am happy to accept that analysis.

In Figure 3A, it's important to note that a 5% IFN γ -producing population is not a large population, suggesting that those T cells could indeed be dysfunctional at the level of cytokine production, but that the difference between G1 and G2 is not in cytokine production potential. The axes in Figure 3B should be slightly modified to demonstrate that the authors are looking at CD28 status in TAA-specific CD4 T cells; the right side axis is fine, but the left side axis is confusing and could lead readers to mistakenly assume that 40% and 60% respectively of THD cells are producing cytokine, which is not the case.

When looking at Figure 4 it's a considerable amount of new data, and the cytokine information substantially strengthens their findings. While cytokine stimulation resulting in proliferation of CD4s would go far in helping to explain the phenotype of the dysfunctional T cells - that data suggests a tolerance phenotype - I leave it to the editor and the authors to decide.

I think that with the new data the text conveys a compelling story, but it would benefit from minor modifications to the text to have consistency in the wording- the authors flip-flop between THD and CD28-, and nonTHD and CD28+ (for example, page 5), and it is a bit more confusing that way than it has to be. One potential fix would be to have CD28 status in the figure legends and THD vs nonTHD in the actual text, but other fixes would also be acceptable. Also consider splitting the second results section into a bit more digestible pieces- it's 1.5 pages long and covers a large range of phenotypic observations. Readers could easily get lost in it.

Provided these small changes, the manuscript is now acceptable for publication.

1. The adviser stated: “I would have liked to see the population of G1 in figure 4 divided into responders and non-responders (or individual coloured dots as the other figures) just to see if there was a difference covered by the non-responders especially in IFN γ response”. If feasible within a reasonable timeframe, please address this comment.

Indeed, we had originally contemplated presenting the data with individual coloured dots so that they would match the rest of the figures. However, the overall end result was a very difficult figure to interpret for the reader. The amount of data and graphs shown in figure 4 is so large that it makes its interpretation hard to grasp unless provided as bar graphs. Nevertheless, we did not observe any significant difference between G1 responders and G1 non-responders. In fact, there are not differences at all with G2 patients (all of them progressors). However, we understand the need to analyse the individual data for those readers who may want to. To address this issue, we have plotted the individual data as suggested by the advisor for global IFN γ responses as an appendix file. In this way, this information would be readily available for the readers.

To clarify this also in the manuscript we added the following sentence (shown underlined here) in page 5 “Interestingly, CD4 T cells from both G1 and G2 patient cohorts were as proficient in IFN- γ , IL4, IL10 and IL2 expression as T cells from healthy donors independently of their CD28 expression (**Fig 4A**) or whether these were T cells from G1 responders or non-responders.”

Reviewer 2.

We have addressed all the minor points raised by Reviewer 2 as follows:

- **In Figure 3A, it is important to note that a 5% IFN γ -producing population is not a large population, suggesting that those T cells could indeed be dysfunctional at the level of cytokine production, but that the difference between G1 and G2 is not cytokine production potential.**

We agree with Reviewer 2 and we have clarified this point in page 5 by added the following underlined sentence: “Indeed, although the average percentages of circulating lung cancer-specific CD4 T cells were low, these did not differ significantly between G1 (responders and non-responders) and G2 patients.”

- **The axes in Figure 3B should be slightly modified to demonstrate that the authors are looking at CD28 status in TAA-specific CD4 T cells; The right side axis is fine, but the left side axis is confusing and could lead readers to mistakenly assume that 40% and 60% respectively of THD cells are producing cytokine, which is not the case.**

We sincerely thank Reviewer 2 for pointing out this. We have modified the left side axis as follows in line with the right side axis: “% Non-THD TAA-specific CD4 T cells”.

- **When looking at Figure 4 it is a considerable amount of new data, and the cytokine information substantially strengthens their findings. While cytokine stimulation resulting in proliferation of CD4s would go far in helping to explain the phenotype of dysfunctional T cells- that data suggests a tolerance phenotype- I leave it to the editor and the authors to decide.**

We sincerely thank Reviewer 2 for the opinion. As mentioned in the previous rebuttal letter, we do have that data as part of a second follow-up study in which we are addressing PD-1 and LAG-3 signals within human T cells. We had considered that this paper already has 7 main figures and 5 additional EV figures, each figure composed of a high number of panels, and that addition of further figures would not help the current manuscript. We do think that cytokine priming data will go nicely in a follow-up study.

I think that with the new data the text conveys a compelling story, but it would benefit from minor modifications to the text to have consistency in the wording-the authors flip-flop between THD and CD28-, and nonTHD and CD28+ (for example, page 5), and it is a bit more confusing that way than it has to be. One potential fix would be to have CD28 status in the figure legends and THD vs nonTHD in the actual text, but other fixes would also be acceptable.

We thank the Reviewer to point out this issue so that we can make the text clearer. We have made an effort to standardize nomenclature within the text so that now it should be more homogeneous and clearer.

Also consider splitting the second results section into a bit more digestible pieces-it's 1.5 pages long and covers a large range of phenotypic observations. Readers could easily get lost in it.

We agree with the Reviewer on this point, especially after including all the cytokine information. To solve this issue, we have split this section into two sections in page 5, by adding an additional section heading as follows: "Absence of cancer-specific CD4 T cells or systemic T cell exhaustion are not behind the lack of objective clinical responses to PD-L1/PD-1 blockade therapies".

Corresponding Author Name: David Escors
Journal Submitted to: EMBO Molecular Medicine
Manuscript Number: EMM-2019-10293